# Magnetically-driven colossal supercurrent enhancement in InAs nanowire Josephson junctions

J. Tiira[1], E. Strambini[1], M. Amado[1,2], S. Roddaro[1], P. San-Jose[3], R. Aguado[3], F.S. Bergeret[4,5], D. Ercolani[1], L. Sorba[1] & F. Giazotto[1]

The Josephson effect is a fundamental quantum phenomenon where a dissipationless supercurrent is introduced in a weak link between two superconducting electrodes by Andreev reflections. The physical details and topology of the junction drastically modify the properties of the supercurrent and a strong enhancement of the critical supercurrent is expected to occur when the topology of the junction allows an emergence of Majorana bound states. Here we report charge transport measurements in mesoscopic Josephson junctions formed by InAs nanowires and Ti/Al superconducting leads. Our main observation is a colossal enhancement of the critical supercurrent induced by an external magnetic field applied perpendicular to the substrate. This striking and anomalous supercurrent enhancement cannot be described by any known conventional phenomenon of Josephson junctions. We consider these results in the context of topological superconductivity, and show that the observed critical supercurrent enhancement is compatible with a magnetic field-induced topological transition.

[1] NEST, Istituto Nanoscienze-CNR and Scuola Normale Superiore, I-56127 Pisa, Italy. [2] Materials Science and Metallurgy, University of Cambridge, Cambridge CB3 OFS, UK. [3] Instituto de Ciencia de Materiales de Madrid, Consejo Superior de Investigaciones Científicas (ICMM-CSIC), Sor Juana Inés de la Cruz 3, 28049 Madrid, Spain. [4] Centro de Física de Materiales (CFM-MPC), Centro Mixto CSIC-UPV/EHU, E-20018 San Sebastian, Spain. [5] Donostia International Physics Center (DIPC), E-20018 San Sebastian, Spain. Correspondence and requests for materials should be addressed to E.S. (email: elia.strambini@sns.it) or to F.G. (email: francesco.giazotto@sns.it).

C oupling a conventional s-wave superconductor (S) to materials based on helical electrons such as topological insulators or semiconductors with strong spin-orbit (SO) interaction like InAs or InSb nanowires (NWs) leads to an unconventional p-wave superconductor. The latter may undergo a topological transition, and become a topological superconductor (TS) hosting exotic edge states with Majorana-like character[1,2]. Most of the early and subsequent experimental efforts to demonstrate these modes have been focused on normal metal-superconductor junctions realized with strong-SO NWs[3-6] with the aim to detect signatures of Majorana bound states (MBSs) emerging for increasing Zeeman fields. Soon after, experiments on Josephson junctions based on helical materials have been performed as well to detect peculiar hallmarks of MBSs in the phase evolution of the Josephson effect, both in the context of topological insulators[7-11] and NWs[12]. Yet, a conclusive evidence of MBSs emerging from these hybrid systems still remains an outstanding experimental goal. This is particularly true for NW-based Josephson weak links where no signatures of TS have been reported in the supercurrent[13-20].

In this work, we investigate the Josephson coupling in Al/InAs-NW/Al hybrid junctions in the presence of an external magnetic field. In particular, we focus on the amplitude of the Josephson critical current $I_C$ which is expected to strongly increase when the topology of the junction enables the emergence of MBSs[21]. Our results are discussed on the basis of the most common and understood phenomena that may affect the Josephson supercurrent as well as alternative unconventional effects specific to the geometry of our experiment. Although no final conclusions can be drawn from such an analysis, we stress that a topological transition appears to be fully compatible from a qualitative point of view with the observed phenomenology: our proposed physical scenario consists of a magnetically driven zero-energy parity crossing of Andreev levels in the junction[21] which is expected to occur for magnetic fields applied perpendicularly to the wire SO axis, exactly as observed in the experiment.

## Results

**Sample fabrication and measurement setup.** Samples are prepared by contacting the NWs with superconducting leads defined by electron beam lithography. Figure 1a shows a scanning electron micrograph of a typical n-InAs-NW-based Josephson junction. The junction's interelectrode spacing $L$ ranges from ~40 nm to ~113 nm. The growth and physical details of the n-doped InAs NWs are reported in the Methods section: Device Fabrication. For each junction a neighbouring Ti/Al superconducting pair is used for transport measurements. The current versus voltage ($IV$) characteristics of the Josephson weak links are obtained by applying a bias current $I$, and measuring the resulting voltage drop across the NW via a room-temperature differential preamplifier, as shown in Fig. 1a. A schematic side view showing the materials forming the junctions as well as a 45° tilted scanning electron micrograph of a typical distribution of n-InAs NWs after the growth are displayed in Fig. 1b,c, respectively.

**Temperature dependence of the supercurrent.** Figure 1d shows the temperature dependence of the $IV$ characteristics of a typical Josephson junction with $L = 100$ nm. A critical current $I_C$ exceeding ~150 nA is observed at the base temperature of a dilution refrigerator (15 mK) (ref. 19). Without any applied magnetic field $I_C$ persists up to ~800 mK. Furthermore, for temperatures below 250 mK a remarkable hysteretic behaviour between the switching ($I_C$) and retrapping ($I_{Cr}$) critical currents is observed, as shown in Fig. 1d,e. This hysteresis stems from

quasiparticle heating within the NW region while switching from the resistive to the dissipationless regime[22], and has been often observed in hybrid Josephson junctions independently of the geometry or composition of the weak link[18,23-27].

The monotonic decay of $I_C(T)$ and the saturation of $I_{Cr}(T)$ at low $T$ is displayed in Fig. 1e (ref. 28) together with the two best fits for $I_C(T)$ obtained by modelling the junction as an ideal diffusive or ballistic NW (red and blue dashed line, respectively). The details of the theoretical model for each fit are described in the Methods section. None of the two fitting curves can accurately describe the monotonic decay of $I_C(T)$ which is consistent with a junction belonging to an intermediate regime between the two above limits[19], that is, $L \sim l_e$ where $l_e \sim 60$ nm is the electron mean free path estimated in our InAs-NWs (refs 20,29. Moreover, the diffusive fit suggests an effective junction length of the order of ~300 nm which largely exceeds the interelectrode spacing. The actual geometry of the junction supports this observation since the superconducting electrodes cover a considerable portion of the NW. In addition, the same fit yields an estimate for the junction Thouless energy $E_{Th} = \hbar D/L^2 \approx 160$ μeV ($D$ is the diffusion constant of the InAs NW), and for the induced superconducting minigap[30] in the NW, $\Delta \simeq 80$ μeV.

**Magnetic field dependence of the supercurrent.** As long as the external magnetic field is absent, the behaviour of our NW-based Josephson junctions is fully consistent with what has been observed for similar Al/InAs-NW/Al weak links[16,17,19,20]. A strikingly different and unexpected phenomenology develops when a magnetic field is applied perpendicular to the substrate ($B_\perp$): the amplitude of $I_C$ drastically changes in a way that has never been observed so far in similar devices[16,19] and, to the best of our knowledge, in any other kind of Josephson weak links. As shown in Fig. 2a, where the $IV$ characteristics of one of the junctions with $L = 100$ nm is displayed for different values of $B_\perp$, the critical current $I_C$ remains almost constant up to ~15 mT, then quickly doubles its amplitude at a switching field $B_{SW} \simeq 23$ mT, and decays at larger magnetic fields. Furthermore, for $|B_\perp| > B_{SW}$, the $IV$ characteristics develop a dissipative behaviour (that is, they show a finite slope around $V \approx 0$, see Fig. 2b) which corresponds to a resistance of ~60 Ω. The colossal enhancement of $I_C$ (which exceeds 100%) is very robust and reproducible over different cooling cycles, and measured junctions.

Figure 2c shows the behaviour of $I_C(B_\perp)$ for three junctions of different lengths. Apart from sample specific fluctuations of $I_C(0)$, the supercurrent enhancement occurs at the same magnetic field for all the junctions. This suggests that the origin of the effect is intrinsic to the materials combination, and cannot be attributed to geometrical resonances in the junction[31]. A nontrivial behaviour characterizes also the temperature evolution of the relative $I_C$ enhancement, $\Delta I_C = \max[I_C(B_\perp)]/I_C(0) - 1$, shown in Fig. 2d,e for two junctions with $L = 40$ nm and 100 nm belonging to different NWs. In contrast to the usual temperature-driven weakening of the Josephson effect, $\Delta I_C$ shows a nonmonotonic behaviour with a maximum at $T \sim 300$ mK for the junction with $L = 100$ nm whereas in the shorter junction ($L = 40$ nm) the behaviour is almost monotonic, and simply decays with the temperature. Several characterizations performed on different samples seem to indicate that the temperature behaviour of $\Delta I_C$ is not related to the length of the junction but rather depends on the specific NW. Moreover, the behaviour of $B_{SW}(T)$ shown in Fig. 2f for a junction with $L = 100$ nm follows the same temperature dependence of the critical field for the disappearance of the Josephson effect $B_C$ (also displayed in the same plot) therefore suggesting a common origin of the two phenomena, that is, the proximity effect.

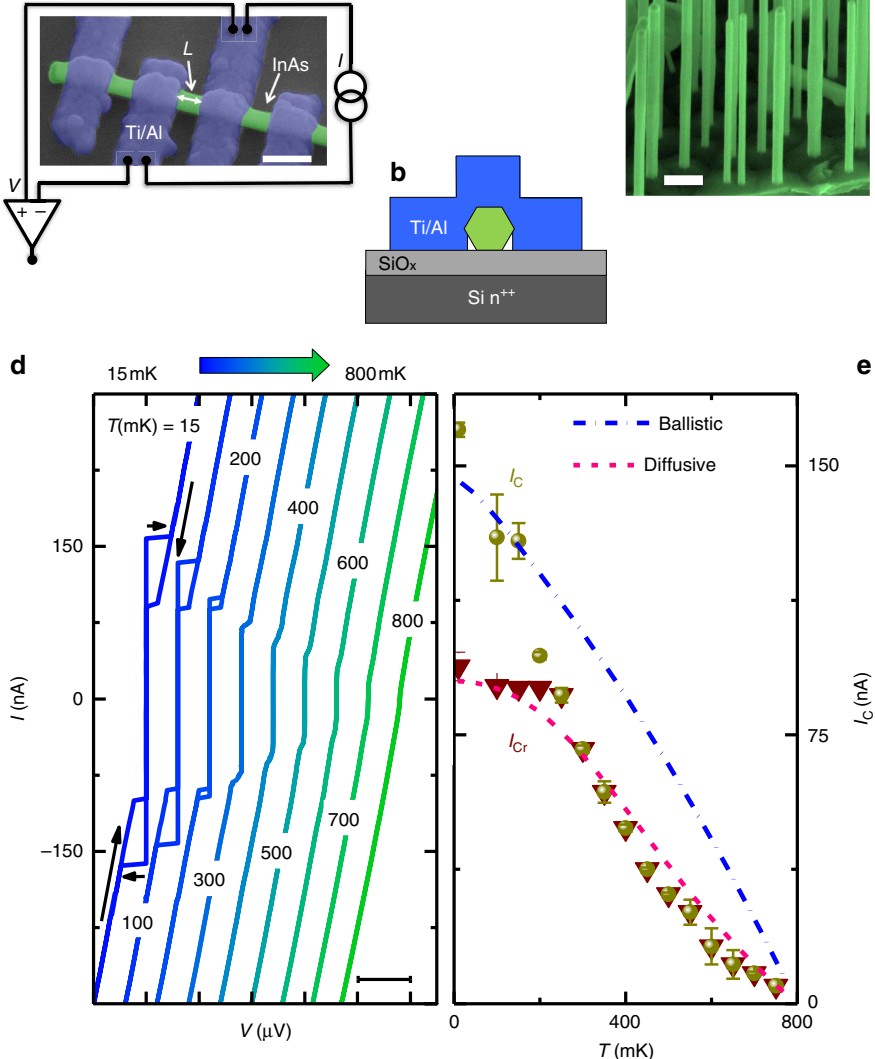

**Figure 1 | Sample layout and zero magnetic field characterization.** (**a**) Pseudo-colour scanning electron micrograph of a typical *n*-InAs nanowire-based Josephson junction along with a sketch of the four-wire measurement setup. The Josephson junction is current biased (*I*) whereas the voltage drop (*V*) is measured through a room-temperature differential preamplifier. The junction length is denoted with *L*, and the width of the Ti/Al electrodes is ∼150 nm as indicated by the scale bar. (**b**) Side view of the junction showing the different materials forming the structure. (**c**) Coloured 45° tilted scanning electron micrograph of a typical distribution of InAs nanowires after their growth, with a scale bar for 200 nm. (**d**) Back and forth current versus voltage characteristics of an Ti/Al-InAs Josephson junction with *L* = 100 nm measured at different bath temperatures *T*. The curves are horizontally offset for clarity. The scale bar of *x* axis is 50 μV. (**e**) Switching (*I*$_C$, dots) and retrapping (*I*$_{Cr}$, triangles) supercurrent versus temperature. The amplitude is estimated from the mean value of the positive and negative critical currents from back and forth branches while the error bar is the difference between these two values. Two distinct theoretical models for the critical current *I*$_C$ holding either in the ballistic (dash-dotted line) or diffusive (dashed line) regime are shown.

A crucial feature observed in the $I_C$ enhancement of all the junctions, and which is essential in order to discriminate over the possible origins of this effect, is the strong dependence of $I_C(B)$ on the orientation of the external magnetic field. This clearly appears in Fig. 3a–c, where we compare the critical currents of two junctions with different lengths ($L = 40$ nm and 113 nm) measured for three different orientations of $B$. The maximum $I_C$ enhancement is observed when the field is applied perpendicular to the substrate (and thus also to the NW axis) as shown in Fig. 3a, and occurs at $B_{SW} \simeq 23$ mT. Differently, in canted field configurations ($B_{30°}$, Fig. 3b) $B_{SW}$ shifts to higher fields according to the amplitude of the projection $B_\perp$. When $B$ is applied in-plane ($B_{\parallel}$, Fig. 3c) the effect is almost absent apart from a tiny supercurrent enhancement around ∼160 mT which can be ascribed to a small misalignment present in

our setup as well as to an incomplete magnetic field screening in the junction region, as will be discussed below. Notably, the in-plane components of the magnetic field (Fig. 3c) seem to have a marginal role in determining the actual value of $B_{SW}$; this conclusion following from the similar behaviour displayed by the two above Josephson weak links which were fabricated with NWs having very different orientations in the substrate plane.

The peculiar dependence of the $I_C$ enhancement as a function of magnetic field direction imposes an important constraint over possible models that can explain the effect. These observations joined with the in-plane pinning of the SO vector in InAs NWs laying on top of a SiO$_2$/Si substrate[3], seem to lead to the intriguing conclusion that the $I_C$ enhancement requires the magnetic field to be perpendicular to

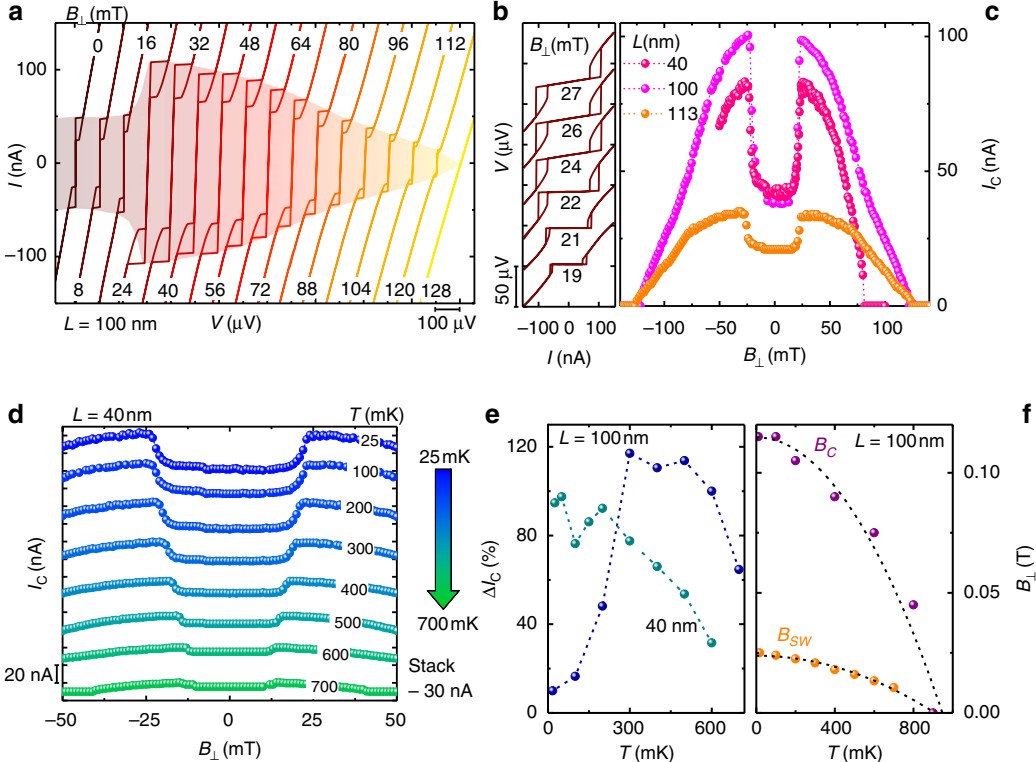

**Figure 2 | Enhancement of the critical current at finite out-of-plane magnetic field.** (**a**) Back and forth current–voltage (*IV*) characteristics of a Josephson junction with $L = 100$ nm measured for different values of the out-of plane magnetic field ($B_\perp$) at 15 mK. The curves are horizontally offset for clarity. A strong enhancement of $I_C$ occurs at $B_\perp \simeq 23$ mT. (**b**) Blow-up of selected *IV* characteristics of the same junction of **a** showing the dissipative behaviour of the weak link after the transition to the enhanced $I_C$ state. (**c**) Comparison of the $I_C$ versus $B_\perp$ behaviour for three junctions of different length *L* at 15 mK. (**d**) Critical current $I_C$ versus $B_\perp$ measured at different bath temperatures for a junction with $L = 40$ nm. The curves are vertically offset for clarity. Each data point represents the critical current obtained from a single *IV* measurement at constant magnetic field and temperature. (**e**) Temperature dependence of the critical current relative enhancement, $\Delta I_C = \max[I_C(B_\perp)]/I_C(0) - 1$, for two different junctions with $L = 100$ nm and $L = 40$ nm. Note that the junction with $L = 100$ nm is different from the one shown in **c**. (**f**) Temperature dependence of the critical field $B_C$ for the disappearance of the Josephson effect, and of the switching field $B_{SW}$ for $I_C$ for a junction with $L = 100$ nm. Dotted lines are the BCS fitting of the two data sets using the equation $B_x(T) = B_x(0)[1 - (T/T_C)^2]$ obtained for the same superconducting critical temperature $T_C = 900$ mK.

the SO vector. The lack of $I_C$ enhancement observed for field configurations orthogonal to the SO vector but parallel to the NW can be attributed to the strong magnetic field expulsion in the weak link region due to the Meissner screening of the superconducting electrodes forming the junction. The magnitude of this effect has been numerically estimated for our junctions geometry for the three relevant directions of the external magnetic field ($B_{ext}$). The results are summarized in Fig. 3d–f where the space distribution of $B$ is calculated assuming an ideal Meissner effect in the superconducting leads. In particular, we obtain that when $B_{ext}$ is orthogonal to the substrate ($B_\perp$, Fig. 3d) the magnetic field $B_{eff}$ in the junction region is strongly amplified, and its intensity is almost doubled with respect to the external field due to magnetic focusing, as recently reported for Pb-based InAs NW Josephson junctions[18]. When $B_{ext}$ is applied in-plane along the SO vector ($B_{SO}$, Fig. 3e) there is complete penetration of the magnetic field within the junction region, that is, $B_{eff} \sim B_{ext}$. By contrast, when $B$ is applied parallel to the NW axis ($B_\parallel$, Fig. 3f) the weak link area is almost screened by the magnetic field thanks to Meissner expulsion in the leads, and $B_{eff}$ obtains values up to $\sim 25\%$ of the external field at the centre of the junction. Figure 3g summarizes the above results for the amplitude profile of $B_{eff}$ along the portion of the NW indicated by the dashed lines in Fig. 3d–f.

## Discussion

The scenario drawn above by the experimental evidence is clear but its interpretation is puzzling due to the complexity of the system and the large amount of non-trivial phenomenologies possible for Josephson junctions. In the first approximation it is known that a magnetic field destroys superconductivity via the orbital and paramagnetic effect, and therefore one would expect a monotonic decay of the critical current by the application of an external field. An experimental exception to this behaviour is represented by field-enhanced superconductivity observed in Josephson junctions made with metallic NWs covered with magnetic impurities. In that case, the $I_C(B)$ enhancement is induced by the field polarization of local moments, and by the relative quenching of the exchange coupling with the electrons in Cooper pairs[32]. Owing to the absence of magnetic impurities in our NWs, and owing to the specific magnetic field orientation leading to the $I_C$ enhancement we can exclude this picture as the explanation of our observations. Moreover from the flat behaviour of the magnetoresistance of the NWs characterized at 2 K ($T > T_C$) we also can exclude the role of the junction resistance in the $I_C$ enhancement. Two further mechanisms are known that yield an $I_C$ increase as a function of field: Fraunhofer-like diffraction[33], and the $\pi$-junction behaviour[34]. The former is expected to occur whenever the magnetic flux enclosed by the junction, $\Phi = LWB_\perp$ (*W* is the NW diameter), equals an integer multiple of the flux quantum, $\Phi_0 = h/2e$. Note that neither the

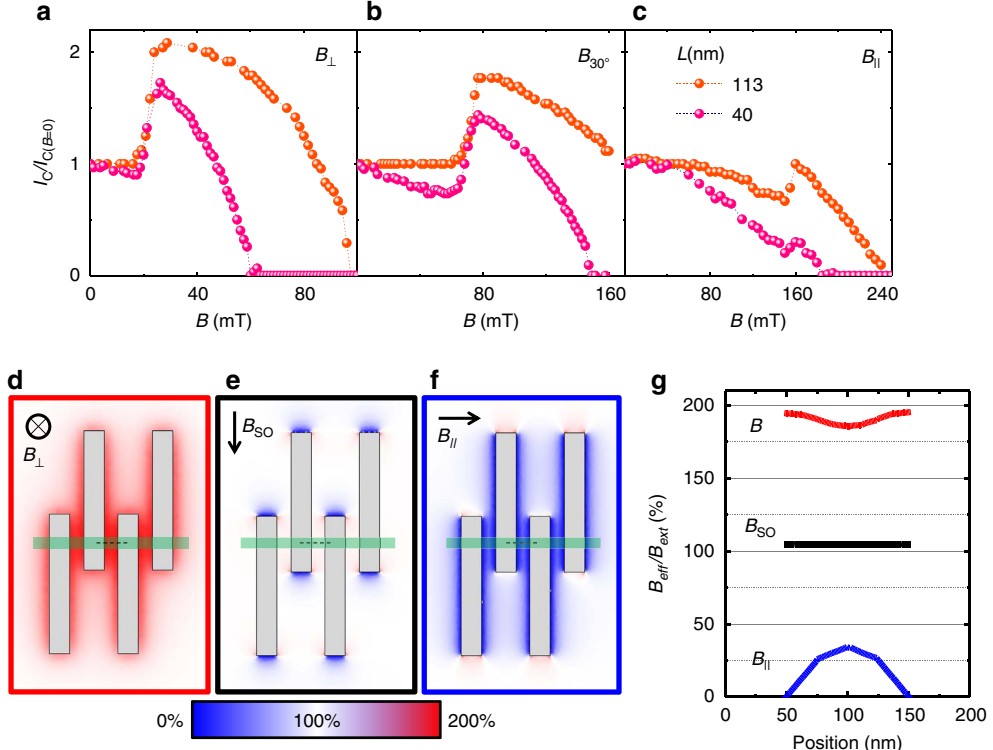

**Figure 3 | Angle dependence of the critical current enhancement.** Comparison between the $I_C$ behaviour as a function of the magnetic field applied in three different orientations for two different junction lengths (**a**) out-of plane ($B_\perp$), (**b**) 30° from the plane ($B_{30}$) and (**c**) in-plane ($B_\parallel$). Finite-element simulation of the nonuniform distribution of the amplitude of $B$ due to the Meissner effect in the superconducting leads, evaluated for a magnetic field applied along the three main orthogonal axes of the junction: (**d**) out-of plane ($B_\perp$), (**e**) in-plane orthogonal to the NW and parallel to the SO vector ($B_{SO}$) and (**f**) along the NW ($B_\parallel$). Vertical grey regions indicate the superconducting electrodes overcoming the light-green horizontal NW. (**g**) Amplitude profile of the effective magnetic field ($B_{eff}$) normalized by the applied magnetic field ($B_{ext}$) along the portion of the NW indicated by the dashed lines shown in **d**–**f** panels. Note that while in the out-of-plane direction ($B_\perp$, red line) the $B_{eff}$ is almost uniformly doubled with respect to $B_{ext}$ owing to the focusing effect, along the NW ($B_\parallel$, blue line) $B_{eff}$ is drastically suppressed by Meissner screening, and reaches at most ~25% of the intensity of the external field at the centre of the junction. In the in-plane direction orthogonal to the NW ($B_{SO}$, black line) $B_{eff}$ almost coincides with $B_{ext}$.

absence of an $I_C$ maximum around $B_\perp = 0$ nor the evidence that the critical current enhancement is independent of the junction length can be explained within this picture. On the other hand, a $\pi$-junction behaviour induced by the external field is typically characterized by oscillations of $I_C(B)$ with periodicity of the order $\sim g\mu_B B/E_{Th}$, that is, the ratio between the Zeeman and the Thouless energy of the junction, where $g$ is the NW gyromagnetic factor and $\mu_B$ is the Bohr magnetron. Such oscillatory behaviour, when combined with SO coupling and disorder[34], might explain enhancement of the critical current for some specific values of the magnetic field. However, neither the estimated value for the ratio $g\mu_B B_{SW}/E_{Th} \sim 0.25$ nor the absence of any dependence on the junction length (through the Thouless energy $E_{Th}$) can be accommodated within this explanation. Enhancement of the critical current as a function of the magnetic field has also been predicted to occur in Josephson junctions with inhomogeneous spin-splitting fields[35–37]. However, according to the theoretical model the field inhomogeneity required to explain the observed enhancement is far from realistic (see Supplementary Note 2 and Supplementary Fig. 3).

We focus here on a last possible scenario based on topological transitions in the ballistic wires which seems to be more suitable for describing our experiment setup. The enhancement of the critical current in a NW-based superconductor-normal metal-superconductor Josephson junction was predicted to occur after the proximitized sections of the NW (the S′ regions shown in Fig. 4a) are driven into a topologically non-trivial phase by an

external Zeeman field. In particular, it was shown[21] that in the topologically non-trivial phases the critical supercurrent of a Josephson junction realized with a multimode quasi-one dimensional semiconductor NW with SO (Rashba-type) coupling, like the ones studied here, can be strongly enhanced relative to the trivial phase for small junction transmissivity ($T_N$). This happens by virtue of the additional supercurrent contributed by Majorana zero modes in the junction as the external Zeeman field exceeds a critical value ($B_{crit}$). In a quasi-one dimensional geometry, this topological transition occurs as the shallowest subband $n$ undergoes a gap inversion at Zeeman energy $V_Z^{(n)} = \sqrt{\Delta^2 + \mu_n^2}$, where $\mu_n$ is the Fermi energy measured from the bottom of the subband, and $\Delta$ is the superconducting minigap induced in the NW. After the transition, and for long enough proximitized regions, that is, for $L_S \gg \xi \equiv \hbar v_F/\Delta$ where $\xi$ is the Majorana state localization length, each S′ section of the NW becomes a TS with emergent Majorana zero modes at its ends (the red circles in Fig. 4a). The topological transition can be directly seen as a closing and reopening of the Boboliubov-De Gennes (BdG) spectrum near zero energy. After the transition, the BdG spectrum contains emergent Majorana zero modes with protected crossings at a superconducting phase difference $\phi = \pi$ which give rise to a $4\pi$-periodic Josephson effect, and to an enhanced critical current $(I_C \sim \sqrt{T_N})$ relative to that of conventional Andreev levels $(I_C \sim T_N)$; therefore, only clearly observable for reduced

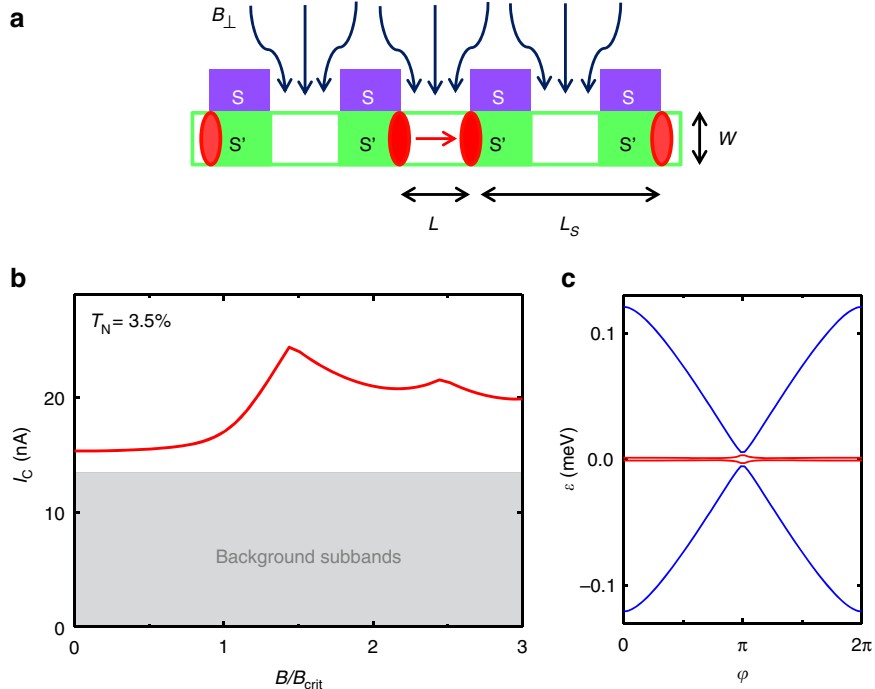

**Figure 4 | Topological model for the $I_C$ enhancement.** (**a**) A sketch of the four Majorana bound states (red circles) formed in the two proximitized regions S′ of a (ballistic) NW under the action of an out-of-plane magnetic field $B_\perp$. (**b**) Theoretical critical current $I_C$ calculated for increasing magnetic field values in an InAs Josephson junction similar to the ones investigated in the experiment ($L = 50$ nm, $L_S = 500$ nm and $W = 40$ nm). The NW charge density $3 \times 10^{18}$ cm$^{-3}$ corresponds to 19 occupied spinful subbands. Rest of the parameters: spin-orbit coupling $\alpha_{SO} = 0.13$ eV Å, contact transmissivity per mode $T_N = 3.5\%$ and zero-temperature superconducting energy gap $\Delta = 200$ μeV. (**c**) corresponding Andreev levels $\varepsilon$ versus phase difference $\phi$ at $B = 1.5B_{crit}$, which exhibit an avoided crossing at phase difference $\phi = \pi$ due to the hybridization of inner (blue) and outer (red) Majorana bound states across $L_S$.

contact transmissivity $T_N \lesssim 0.5$. For shorter wires (with $L_S \ll \xi$) the Majorana modes are always strongly overlapping, and merge into standard finite-energy Andreev levels. We show below, however, that these states are still able to sustain a sizeable critical current increase at $V_Z > V_Z^{(n)}$ despite the overlap, particularly if $\mu_n$ is small. The reason is that, in this case, their corresponding hybridization away from zero energy is very small (their splitting scales with the Fermi wave vector[38,39], which reaches its minimum $k_F \sim 2m\alpha_{SO}/\hbar^2$ for $\mu_n = 0$). The overlapping Majoranas thus remain close-to-zero modes, and effectively behave as Majorana precursors, revealing through $I_C$ the underlying topological transition.

We compute, using the method of ref. 21, the critical current across an NW Josephson junction, using a finite $L_S = 500$ nm as corresponds to the experimental samples (note that we take into account all the S fingers on each side of the junction). Given the charge density of the NWs $3 \times 10^{18}$ cm$^{-3}$, and assuming a hexagonal NW section with a face-to-face distance of $\sim 37$ nm, we calculated that a total of $N \approx 19$ spinful subbands are populated at $V_Z = 0$ (see Methods: numerical simulation of the 1D subband edges). The shallowest subband is found to be almost depleted, so that $\mu_n \approx 0$. Assuming an induced gap $\Delta \approx 250$ μeV, and an average normal conductance of $T_N \approx 3.5\%$, these 19 modes contribute to a total critical current of approximately $I_C = NT_N e\Delta/2\hbar \approx 20$ nA at $V_Z = 0$ (estimated in the Andreev approxmation $\mu_m \gg \Delta$ for each mode $m$), or 21 nA when computed exactly in our model. This value matches our observed $\sim 21$ nA for the longer junctions ($L = 113$ nm, orange dots in Fig. 2c). As $V_Z$ exceeds $V_Z^{(n)} \approx \Delta$, the shallowest subband becomes inverted, and its contribution to $I_C$ is expected, in the Andreev approximation, to increase from $I_C^{(n)} \approx T_N e\Delta/2\hbar$ at $V_Z = 0$ to around $I_C^{(n)} \approx \sqrt{T_N}e\Delta/2\hbar$ due to the presence of the

Majorana precursors, an enhancement of $\sim 4$ nA. This is actually an underestimation of the $I_C$ enhancement in our case, for which the Andreev approximation does not hold. A precise computation using the approach of ref. 21 yields a supercurrent increase of more than two times larger, around 10 nA, again in rather good agreement with values observed for the longer junctions. The large increase in $I_C^{(n)}$ stems mostly from the change from $T_N$ to $\sqrt{T_N}$, a signature of Majorana precursors. Quantitatively, this effect is further amplified by an increase in $T_N$ for the shallowest subband as a function of $V_Z$. Owing to the small $\mu_n$, the Fermi momentum is small at $V_Z = 0$ (and thus also $T_N$), but is enhanced as $V_Z$ increases, which in turn strongly enhances $T_N$.

Figure 4b,c illustrate the critical current $I_C$ from the above simulation and the calculated Andreev spectrum, corresponding to parameters relevant to our long junctions. Note the sharp $\sim 50\%$ enhancement around the topological transition. We emphasize that, due to the finite $L_S$, the junction is technically topologically trivial since the protected $\phi = \pi$ crossing is slightly lifted, as shown in Fig. 4c. Despite this, the lifting is small, and consequently, the $I_C$ enhancement predicted to occur at the topological transition for ideal junction lengths ($L_S \gg \xi$) remains clearly visible, even in our short-$L_S$ geometry ($L_S \sim \xi$). This shows that Majorana precursors in finite length junctions contribute strongly to $I_C$ despite their overlap.

Two issues remain in the above topological interpretation. The first is that shorter junctions exhibit $I_C$ enhancements that are around twice that of longer junctions, and exceed the maximum supercurrent $e\Delta/2\hbar \sim 30$ nA that may be carried by a single Majorana mode. This could be due to subband pairs becoming inverted simultaneously. Such scenario becomes possible if the NW charge density is slightly increased, for example, by charge transfer from the superconductor, thus

pushing the Fermi energy close to a subband doublet, as those shown in Supplementary Fig. 2.

Subband doublets are weakly coupled modes with opposite angular momentum. Pairs of Majorana precursors from subband doublets will contribute independently to $I_C$ as the interband mixing due to SO coupling is negligible for our NW widths (they are in the so-called approximate BDI symmetry class), thus doubling the $I_C$ enhancement. A second issue is the precise value of the critical field $B_{crit}$. There is considerable uncertainty in the system parameters involved, but using the bare $g$-factor $g = 15$ for InAs and $\Delta = 250\,\mu eV$, and assuming ideal proximity effect of the Al leads, we estimate $B_{crit} = \Delta/((1/2)g\mu_B) \simeq 580$ mT. This value is quite larger than the observed critical field $B_{SW} \approx 23$ mT. The discrepancy may be explained by a combination of factors, including the possibility of a strong enhancement in the $g$-factor due to carrier confinement (recently $g$ factors up to 50 have been observed in similar NWs[6]), a smaller value for the induced gap $\Delta$ than assumed here due to a non ideal transparency of the Al/NW interfaces, or pairing suppression induced by the Zeeman field[40]. We note moreover that, among all the obvious energy scales in the problem, $V_Z^{(n)} \approx \Delta$ remains the smallest, and therefore the most likely to be ultimately involved in the transition observed at $B \sim 23$ mT.

Finally, it is interesting to note that the small $I_C$ enhancement observed in Pb-based InAs NW Josephson junctions[18] and described in terms of a Fraunhofer pattern is also compatible with the same topological transition appearing at the switching field expected for the higher Pb pairing potential. Despite the open issues, the topological interpretation of our experiment is appealing for one further reason. It quite naturally explains why a dissipative component appears in transport concurrent with the supercurrent enhancement. Since in the topologically non-trivial phase the superconducting gap of the inverted mode is $p$-wave like, it is not protected against disorder by the Anderson theorem, and as a result is likely to become smoothed in actual samples, with finite, disorder-induced subgap quasiparticle weight, which precludes a strictly dissipationless supercurrent[41]. This is a known issue in other experimental efforts towards topological superconductivity such as Rashba NWs[3,42], where the induced gaps typically soften as magnetic field increases[43].

In summary, we have reported a colossal enhancement of the critical supercurrent at finite magnetic fields in $n$-InAs NW-based weak links which is in total contrast with the behaviour observed so far in conventional Josephson junctions. The effect manifests itself only for very specific experimental conditions: the magnetic field needs to be applied perpendicular to the NW axis and to the substrate, that is, in the only configuration which is expected to be perpendicular to the SO vector in the weak link. This is in stark contrast with what has been observed so far in conventional weak links, where the Josephson coupling turns out to be suppressed by any applied magnetic field. Notably, this abrupt switching of $I_C$ always occurs around the same magnetic field ($B_{SW}$) in junctions made of nominally identical NWs but characterized by different lengths of the N region, therefore suggesting that the origin of the observed critical current enhancement is intrinsic, that is, it is not related to geometrical resonances existing in the junction or linked to the Thouless energy of the system[31]. In addition, the temperature dependence of $B_{SW}$ follows a BCS-like behaviour thus indicating a strong link to the proximity effect present in the junction. Despite this clear but puzzling experimental evidence, a conclusive theory capable to fully describe our results is still missing.

We presented a model, based on topological transitions, that allows us to qualitatively explain all the observed phenomena. Quantitatively our theoretical prediction gives only a rough estimate of the transition field $B_{crit}$, still one order of magnitude

higher than the experimental one $B_{SW}$. This discrepancy might be reduced by incorporating non-trivial but experimentally relevant effects into the theory, including finite size in the NW, electrostatic effects and scattering events. Yet, the realization of fully ballistic junctions in which the carrier density can be controlled by means of an additional gate is expected to provide an improved understanding of the present phenomenology, thanks to the fine tuning of the NW energy levels. In addition, extension of the theory of topologically non-trivial pairing to the case of diffusive SO-NW Josephson weak links will offer a refined description of our system, and further clarify the role of MBSs in the experiment. We believe that these newsworthy results will stimulate the development of novel theoretical models, and will contribute to the progress of the investigation and understanding of MBSs in condensed matter physics.

## Methods

**Device fabrication.** The Josephson junctions presented in this work are based on heavily $n$-doped InAs NWs grown by metal-assisted chemical beam epitaxy[44]. NWs are grown in a Riber C-21 reactor by using metallic seeds obtained from thermal dewetting of a thin Au film layer evaporated on a InAs substrate[18,45]. Trimethylindium (TMIn) and tertiarybutylarsine (TBAs) (cracked at 1,000 °C) are used in the growth as metalorganic precursors while ditertiarybutyl selenide (DtBSe) is used as a selenium source for $n$-type doping. Based on previous experiments performed on similar NWs[18,29] we estimate a carrier density $n_s \sim 3 \times 10^{18}\,cm^{-3}$ and an electron mobility $\mu \sim 2,000\,cm^2\,Vs^{-1}$ from which we deduce a Fermi velocity $v_F \approx 2.2 \times 10^6\,m\,s^{-1}$, an electron mean free path $l_e \sim 60$ nm and the effective electron mass for InAs NWs $m^* = 0.023m_e$, where $m_e$ is the free-electron mass.

After the growth, the NWs are transferred mechanically onto a $SiO_2$/$n$-Si commercial substrate pre-patterned with Ti/Au pads and alignment markers which are defined by optical and electron beam lithography, and deposited by thermal evaporation. The position of the NWs on the substrate is mapped with a scanning electron microscope, and used for the aligned electron beam lithography of the Josephson junctions. Before the deposition of the Ti/Al superconducting electrodes the NWs are etched with a highly diluted ammonium polysulfide $(NH_4)_2S_x$ solution to remove the native oxide layer present on the semiconductor surface. This procedure improves the quality of the ohmic contact, limiting undesired surface scattering processes. The deposition of the Ti/Al (12/78 nm) leads is performed at room temperature in ultra-high vacuum conditions by electron beam evaporation[18]. More than ten junctions were fabricated starting from five different NWs, and measured at low temperature.

The magneto-electric characterization of the InAs-NW Josephson junctions was performed in a filtered dilution refrigerator down to 15 mK using a standard 4-wire technique. The current-voltage characteristics of the junctions were obtained by applying a low-noise biasing current, with voltage across the NW being measured by a room-temperature battery-powered differential preamplifier.

**Fitting details of $I_C(T)$.** To study the decay of $I_C(T)$ presented in Fig. 1e and identify the main transport regime holding in the NW we have modelled the Josephson junction in two opposite limits: diffusive and ballistic. In the diffusive regime, $I_C(T)$ is fitted with the expression of the critical current of a superconductor-normal metal-superconductor (SNS) junction obtained by solving the linearized Usadel equation[46]

$$I_C = \frac{\pi k_B T}{e R_{NW}} \sum_\omega \frac{(\kappa_\omega L)\Delta^2}{(\omega^2 + \Delta^2)[\alpha \sinh(\kappa_\omega L) + \beta \cosh(\kappa_\omega L)]}, \quad (1)$$

where the sum is over the Matsubara frequencies $\omega = \pi k_B T(2n+1)$, $n = 0, \pm 1, \pm 2, ...$, $k_B$ is the Boltzmann constant, $\kappa_\omega = \sqrt{2|\omega|/(\hbar D)}$, $D$ is the NW diffusion coefficient, $\hbar$ is the reduced Planck constant, $R_{NW}$ is the resistance of the NW of length $L$, $\alpha = 1 + r^2(\kappa_\omega L)^2$, $\beta = 2r(\kappa_\omega L)$, and $r = R_b/R_{NW}$ with $R_b$ being the resistance of the SN interface. The best fit with the diffusive model presented in Fig. 1e (pink dashed line) is obtained from equation (1) by using $L = 300$ nm, $D = 0.0416\,m^2\,s^{-1}$, $R_b = 4\,\Omega$ and $R_{NW} = 741\,\Omega$.

On the other hand, for the fit in the ballistic regime we use the expression of the Josephson current $I_J$ valid for a multichannel junction[47]

$$I_J(\phi) = \frac{e\Delta(T)^2}{2\hbar} \sin\phi \sum_{n=1}^{N} \frac{D_n}{E_n} \tanh\left[\frac{E_n(T)}{2k_B T}\right], \quad (2)$$

where $D_n$ are the eigenvalues of the transmission matrix describing the junction, $N$ is the number of conducting channels, $E_n(T) = \Delta(T)\sqrt{1 - D_n \sin^2\phi/2}$, $\Delta(T)$ is the temperature-dependent BCS energy gap, and $\phi$ is the macroscopic quantum phase difference over the junction. The critical current at each temperature is then obtained by maximizing $I_J(\phi)$ with respect to $\phi$, $I_C = \max_\phi|I_J(\phi)|$. The best fit

with the ballistic model shown in Fig. 1e (blue dashed line) is obtained by setting $\Delta_0 = 120\,\mu eV$, $D_n = 1$, and $N = 5$.

**Model for the $I_C(B)$ enhancement by a topological transition.** A proximitized two-dimensional Rashba semiconductor may be modelled by the Bogoliubov-de Gennes Hamiltonian

$$H = \left(\frac{p^2}{2m^*} - \mu\right)\tau_z + \frac{\alpha_{SO}}{\hbar}\left(\sigma_y p_x \tau_z - \sigma_x p_y\right) \\ + \Delta\sigma_y\tau_y + V_Z\sigma_x\tau_z, \tag{3}$$

where $\sigma_i$ are the Pauli matrices in the spin sector, $\tau_i$ are Pauli matrices in the electron-hole sector, $p^2 = p_x^2 + p_y^2$, $m^*$ is the effective mass, and $\alpha_{SO}$ is the Rashba SO coupling. The last two terms in $H$ describe a superconducting s-wave pairing of strength $\Delta$ induced on the semiconductor and a Zeeman splitting $V_Z$ produced by an external magnetic field. The s-wave pairing $\Delta$ translates into both an effective $p_x \pm ip_y$ intraband pairing, and an interband s-wave pairing when projected onto the basis of $\pm$ eigenstates of the helical Rashba + Zeeman normal problem[1,2]. When the Zeeman energy $V_Z$ exceeds a critical value, the system develops only one of the two p-wave pairings. At that moment it becomes topologically non-trivial.

In a quasi-1D NW geometry, the transverse momentum $p_y$ becomes quantized and discrete confinement subbands develop. In such case there is a critical $V_Z$ for each subband, which reads

$$V_Z^{(n)} = \sqrt{\Delta^2 + \mu_n^2}. \tag{4}$$

Here $\mu_n$ the Fermi energy measured from the bottom of the subband. When subbands are not coupled by a transverse SO coupling (BDI symmetry class), each one leads to Majorana zero modes at either end of the NW. If SO does couple subbands (the so-called D symmetry class, relevant for NW widths comparable to or exceeding the SO length), an even number $N$ of Majorana zero modes at each edge hybridize and form $N/2$ full fermions (standard Andreev levels) at finite energy. If the number of Majoranas is originally odd a single Majorana remains at zero energy in the D class, which is then topologically non-trivial. Otherwise one has a trivial D class system.

This phenomenon leads to a spectral even–odd effect as $V_Z$ is increased and the D-class NW alternates between trivial and non-trivial. The corresponding appearance and disappearance of zero-energy Majorana modes is reflected in the Josephson effect. In particular, the additional supercurrent contributed by the unpaired Majorana zero modes at the junction allows one to directly map the D class topological phase diagram of proximitized multiband quasi-one dimensional semiconducting NWs, as demonstrated in ref. 21.

**Numerical simulation of the 1D subband edges.** The theoretical models proposed in the main text rely on an estimate of the one-dimensional subband filling and spacing for the studied semiconductor NWs. InAs wires grown on the 111 facets display an atomically perfect hexagonal cross-section therefore were numerically simulated using an hexagonal hard-wall confinement. This choice is motivated by the fact that the pinning of the Fermi potential in InAs occurs in proximity of the conduction band edge. As a consequence, InAs does not display surface depletion as generally observed in most of semiconductors but, rather, a surface charge accumulation. The exact band bending along the NW radial direction is in principle non-trivial and it is caused by the combined action of the dopants and of free electrons confined in the various populated one-dimensional subbands. Provided the relatively large doping of our NWs, we do not expect strong deviations from local charge neutrality and the subband eigenmodes were thus calculated assuming a flat band condition in the NW body. Numerical calculations were performed using the commercial PDE solver: COMSOL Multiphysics v.4.2 (COMSOL Inc., 2011).

Schröedinger equation was solved over a hexagonal domain with boundary conditions $\Psi = 0$, to simulate an infinite hard-wall surface confinement. No potential energy term was included, to reproduce the flatband condition. The face-to-face size of the hexagon was set to 35 nm and eigenstates were calculated using a mesh size of about 0.5 nm. An example of the used mesh is visible in Supplementary Fig. 1a, while the corresponding eigenstate number 19 is reported in Supplementary Fig. 1b.

Calculated transverse modes closely resemble, most of the times, the ones that can be analytically calculated for a circular geometry with hard-wall boundary conditions. For instance, the solution reported in Supplementary Fig. 2 clearly resembles the circular state with one radial mode and azimuthal angular momentum. In the same figure we report a direct comparison between the numerically calculated eigenstates for an hexagonal confinement (black hexagons) and those obtained analytically for an infinite hard-wall circular confinement (red circles). Deviations between the two solutions are small for the first modes but start to become more and more relevant for states with angular momenta which are able to sample the hexagon. For instance, the $\pm 3$ modes (eigenstates number 7 and 8) leads to the splitting between degenerate modes number 7 and 8. Fermi energy is then calculated to match, within the Hexagonal confinement, the charge density of the NW ($3 \times 10^{18}\,cm^{-3}$). Notably the Fermi level lies very close to the last occupied subband responsible of the topological transition in our model.

**Data availability.** The data that support the findings of this study is available from the authors on a reasonable request.

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

## Acknowledgements

Partial financial support from the Marie Curie Initial Training Action (ITN) Q-NET 264034, from the MIUR-FIRB2013 Project Coca (Grant No. RBFR1379UX), from CNR through the bilateral CNR-RFBR project 2015–2017, and from the European Research Council under the European Union's Seventh Framework Programme (FP7/2007–2013)/ERC Grant 615187-COMANCHE is acknowledged. The work of E.S. is funded by the Marie Curie Individual Fellowship MSCA-IFEF-ST No. 660532-SupeMag. M.A. acknowledges the support of the Tuscany region through the project 'TERASQUID' and the People Programme (Marie Curie Actions) of the European Union H2020 Programme under REA grant agreement n [656485]. The work by P.S-J. is supported by the Spanish Ministry of Economy and Innovation through Grant No. FIS2015-65706-P and the Ramón y Cajal programme. The work of R.A. is funded by the Spanish Ministerio de Economía y Competitividad through grant FIS2012-33521. The work of F.S.B. is supported by the Spanish Ministerio de Economía y Competitividad through the Project No. FIS2014-55987-P and Grupos Consolidados UPV/EHU del Gobierno Vasco (Grant No. IT-756-13).

## Author contributions

J.T. fabricated all the samples and performed the first measurements with M.A.. J.T., E.S. and F.G. analyzed the measurement data, and worked to find a theoretical explanation. S.R. simulated the magnetic field focusing effects in the system. F.S.B. provided the theoretical model for the supercurrent enhancement induced by an inhomogeneous Zeeman field. R.A. and P.S.-J. conceived the theoretical model based on topological transitions. D.E. and L.S. provided the nanowires used in this work. All the authors were involved in writing the manuscript.

## Additional information

**Competing interests:** The authors declare no competing interests.

**Publisher's note**: 

