## [Peer Review File · Nature Communications]

Reviewers' comments:

Reviewer #1 (Remarks to the Author):

The manuscript, by Paajaste et al., presents intriguing behaviour of critical currents in Al/InAs nanowire SNS junctions under an applied magnetic field. The experimental data and device characterization is laid out very clearly in figures 1-3 and the accompanying text. The critical current and hysteresis behaviour is indeed very similar to what has been reported in several previous papers by other groups. The new effect, a large jump in I_c that happens at a very similar field value in 3 different devices, certainly has not been reported on before, and does strongly compel one to ask what's happening. The authors reasonably argue that the effect is intrinsic, and not dependent on detailed device geometry. Further, the peculiar dependence on field orientation - with no (or very little) effect when the field is aligned with the spin-orbit vector (at least, reasonably assumed to be the SO direction) - does suggest a topological transition as a possible mechanism. If that were the case, the paper would indeed be a very interesting and worthy contribution.

After laying out all the experimental details, the authors turn to possible theoretical explanations - (1) a transition to a topological regime, (2) an exotic effect due to inhomogeneity of Zeeman splitting. This is the least convincing part of the paper, but a crucial one, and careful consideration raises a number of important questions which I believe the authors must address before this paper is further considered for publication.

1) A major criticism is that both theoretical models require very low transmittance to qualitatively explain the observations, whereas all experimental indications point to the opposite regime. The fact that supercurrents up to 150 nA are observed suggests relatively high interface transparency, and the semi-ballistic transport in the nanowire region means that overall transmission should be much higher than the order of 1% invoked to match the topological transition model. A $\sim 1\%$ transmittance would correspond to a tunnel junction, not an SNS junction, in my opinion. In fact, this inconsistency is probably large enough to comfortably rule out this particular explanation. Is there another mechanism by which the topological phase transition could increase I_c , but in the regime of high transmittance?

Corollary 1: the authors could easily estimate interface transparency using the so-called excess current (by measuring the I-V to higher biases) - this should be part of the basic device characterization.

Corollary 2: it's a stretch to say that either model even qualitatively fits the observation; as field is increased I_c jumps fairly abruptly, then slowly decreases monotonically. Model 1 does not give an abrupt jump, in fact the major increase in I_c occurs well above B_{crit} (at a factor of 2 or 3 higher at least). For model 2, again there is not an abrupt jump, and it looks as if the I_c will go very quickly to zero after the peak (slope is incredibly steep).

2) The authors effectively rule out model 2 (inhomogeneous Zeeman field effect) because the inhomogeneity cannot be as large as the model would require, and they say as much in the abstract; however, it actually isn't made clear in the paper itself that they are ruling this out (which is confusing). I think it is useful and important that they show why this can be ruled out, but perhaps given that ultimately it is not a viable explanation, perhaps this could be included in Supplementary material rather than in the main paper.

3) Another major criticism is that it's not clear whether the authors see this effect only in the central junction or not. The picture of Fig 4a suggests that the topological model can only explain the data with this rather ad hoc assumption that the outer junctions behave very differently from the central

junction. I don't see any justification from the device geometry to support this. Do the authors have transport data to back this up? Presumably they can also measure the I-V's of the outer junctions, and report whether the same overall SNS behaviour is observed, and indeed whether this phenomenon (jump in I_c) is observed. If those junctions show similar behaviour, it would appear to rule out the picture of Fig 4a. Certainly, it would go a long way to clarifying the paper if the authors showed the I-V characteristics of all three junctions on one nanowire for comparison. It is mentioned that 10 junctions were fabricated, but data from only 3 are shown...can the authors tell us how many of the junctions worked and also showed this same critical current behaviour?

4) Gating of the nanowire is not discussed, even though it is fabricated on n++ Si and therefore has a back gate. I assume this is because the nanowires are heavily doped and so the gate is ineffective. Is this correct? If the devices can be gated at all, it would have been very interesting to see the dependence of this effect on changing the chemical potential. For example, the even-odd effect with respect to number of occupied subbands could be tested.

5) The authors should measure the normal state conductance (e.g. at a temperature just above the SC transition temperature) versus field to rule out some other magnetoconductance effect unrelated to superconductivity. I agree it is unlikely, but to be sure this should be checked.

To summarize: I believe the authors have observed an intriguing effect, and have done a fairly convincing job to show that it is likely an intrinsic, non-trivial effect. However I believe there are simple things they can do that will go a long way to clarifying what's really happening: estimating contact transparency from I-V data, reporting on all junctions in a single device, measuring gate dependence (if this is possible), and checking normal state conductance versus field. As it stands, the models proposed appear to be unsatisfactory for explaining, even qualitatively, the observed behaviour. For these reasons, I recommend the manuscript not be published in its current state.

Reviewer #2 (Remarks to the Author):

This paper reports measurements of critical currents in Josephson junctions made of InAs nanowires as a weak link. The authors identify a strong enhancement of the critical current when a magnetic field is applied and reaches a critical value, that depends on the magnetic field orientation. They examine the dependence of the effect on several parameters (magnetic field direction, geometrical dimensions, temperature) and assess the relevance of various scenarios. They conclude that « the observed I_c enhancement is compatible with a magnetic field induced topological transition of the junction ».

In general, the measurements are clean. The study of several relevant parameters is thoroughly carried out and several devices have been studied, giving a systematic character to the observations. In consequence, the experimental evidence for the sudden enhancement of the critical current is convincing.

The authors also conduct a study of different theoretical explanations but however fail in giving very strong arguments in favor of the occurrence of a topological transition in their system. The following points in that arose during my reading of the manuscript. Points 1), 2) and 3) are of particular importance.

1) Providing microscopic information on the device is necessary to justify the relevance of the topological transition of Ref.20. For example, this theory applies mainly to a low transmission regime while the authors do not show particular indications that this regime is applicable here. Besides, though the authors qualify the effect of « colossal », the amplitude of the critical current enhancement

is not used as an argument to substantiate any of the presented scenarii. Having estimated the induced gap and the electron density, I would imagine that the authors could evaluate the number of modes, their average transmission, and comment on the amplitude of the change in I_c . It is a rather important point to address, in order to justify how the appearance of a limited number of modes results in a massive doubling of the critical current. Could the authors comment on that?

2) The properties of superconducting thin films such as the ones used in this study can be particularly complicated in a magnetic field. In particular, the film has to overcome the non-negligible height of the nanowire, so that a « perpendicular » magnetic field is never totally normal to the superconducting surface. Do the simulations of Fig.3b take this effect into account ? Have the authors established experimentally the behavior of their films via studies of its resistance or of its diamagnetic properties ?

3) As noticed by the authors, the I-V characteristics reveal the appearance of a residual dissipative behavior in the supercurrent (with a resistance of approximately 60Ω). No possible reasons is given for this surprising feature in the article, which in fact even questions the designation « supercurrent ». Can the authors speculate on origins for the observed phenomenon ? At this stage, it is difficult for me to exclude quenching of superconductivity in the narrow Al superconducting leads, as studied for example in Vercruyssen et al., PRB 85, 224503. Especially Figs. 6 and 7 are quite similar to Fig. 2b of this paper. Together with 2), one could at first sight argue that the supercurrent in fact collapses (no enhancement) and the system then gets into an intermediate state of resistance 60Ω . Can the authors exclude this behavior (from measurements of the leads resistance for instance) ?

4) In a related note, the authors have previously demonstrated the possibility to use Pb contacts to realize Josephson junctions on InAs nanowires. But as far as I understand, these devices do not exhibit the same enhancement of the critical current when a magnetic field is applied, but a more conventional Fraunhofer pattern. Could the authors comment on this fact, and on the advantage of using Al ?

5) It has been proposed (in particular by some authors of the present paper) to examine the threshold for finite voltage features such as the excess current or multiple Andreev reflections. This onset should indeed be observed at Δ/e in the presence of Majorana Bound States (and not $2\Delta/e$ in the conventional case). Have the authors tried to establish the presence of an excess current and to observe changes in its onset around the observed transition ?

To conclude, the paper is well written and relies on clean experimental data. A clear demonstration of a topological transition in InAs nanowires would in my opinion qualify for a publication in Nature Communications, but such a claim is insufficiently justified in the current state of the manuscript. Crucial information is still needed to support the authors' interpretation. The authors could possibly make use of the previously raised issues to further confirm (or infirm) the hypothesis of a topological transition.

Reviewer #3 (Remarks to the Author):

In this manuscript the authors report the observation of a large (up to about 100%) enhancement of the critical supercurrent at finite magnetic fields in Josephson junctions based on semiconductor nanowires proximity coupled to s-wave superconductors. The system under consideration has all the key ingredients that are required for the realization of topological superconductivity (and the corresponding Majorana bound states) in solid state hybrid structures: strong spin-orbit coupling, (proximity-induced) superconductivity, and magnetic field (to break time-reversal symmetry). From

this perspective, the work described here fits well into the larger ongoing effort to understand the physics of hybrid structures that may support nontrivial topological phases and exotic quasiparticle excitations. Moreover, in my opinion, the main finding of this work is very interesting and rather puzzling. On the other hand, I find the possible explanations of the observed phenomenon to be unconvincing and potentially misleading.

I recognize that the system is complex and that modeling it theoretically represents a difficult task. However, the scenario suggested by the authors as being the most likely (enhancement generated by the system undergoing a magnetic field-induced topological phase transition) is not very well supported by the experimental evidence. Here are some of the aspects that I find troubling.

1) Dependence on the orientation of the magnetic field. Most of the signatures consistent with the presence of Majorana bound states (MBSs) that were observed so far were obtained with a magnetic field oriented parallel to the wire. I do not see a convincing reason for having MBSs for a field oriented perpendicular to the substrate and not having them for a field parallel to the wire. The magnetic focusing effect may be responsible for some quantitative differences, but it cannot change the physics qualitatively. On the other hand, I can see a strong reason for NOT having MBSs in the perpendicular geometry. Basically, the Zeeman field under the Ti/Al superconductor (where induced topological superconductivity is supposed to emerge) is very small and it would be extremely surprising to find that it satisfies the topological condition. Also, assuming that the structure from Fig. 1a can be (qualitatively) modeled by the structure shown in Fig. 4a is a stretch. Why would the uncovered middle segment be any different from the other two segments? If they are not, there should be four pairs of strongly overlapping MBSs (instead of two).

2) Short length wires. The superconducting segments of the wire are very short; one cannot talk about a topological phase transition in these conditions. In the best case scenario one can have some strongly overlapping bound states that can be interpreted as "precursors" of the MBSs, but the emergence of these states (e.g., the "critical" field associated with their occurrence and the field range over which they are present) depend strongly on the details of the system.

3) Other problems. The estimated induced gap is not consistent with the observed critical field. There is a factor of 8 between the two quantities; even considering a focusing factor of 2 the difference remains significant. Moreover, applying the factor of 2 uniformly is wrong. If a state extends throughout the whole L_s region, the effective Zeeman field that it "feels" is closer to the bare field (not the focused one). If, on the other hand, we consider a state localized right outside the superconductor, then it has nothing to do with the topological phase transition (that characterizes the superconducting segment of the wire, which has a much lower Zeeman field).

In conclusion, the manuscript describes an interesting phenomenon, but provides no convincing explanation for the observed physics. In addition, the study could have been more thorough; a straightforward test that could provide further support for the topological transition scenario (or eliminate it) would have involved longer superconducting regions and (possibly) partially covered wires (with a magnetic field oriented along the wire). In the absence of these tests, the work presented in the manuscript poses some interesting questions, but does not clarify the outstanding problems already existing in this field.

Reviewers' comments:

Reviewer #1 (Remarks to the Author):

I've carefully reviewed the new material, and find the replies to my queries are sufficient and the revised manuscript much improved. The more complete characterization of the devices in terms of number of occupied subbands together with a more convincing theoretical model make the topological explanation much more plausible. In particular, the explanation that I_c scales with the square root of the transmission for the zero modes, and that transmission is of order 5%, resolves one of main problems I had with the paper. The only worrisome thing is the remaining discrepancy between the predicted and observed critical fields. The authors note that this could be explained by a combination of factors, such as increased g-factor due to confinement and non-ideal proximity gap. Do they also mean to include the field focussing effect? That would give a factor of two, which would mean the other factors only need give a factor ~ 10 , which is more plausible than a factor 20. Despite this issue, I think overall their picture is a good candidate to explain the observations. I can't say with 100% certainty that it is the correct explanation, but this new version lays out a much more compelling argument than the first version. I would now support publication. I think it likely that the topological explanation is correct. However, even in the (unlikely) event that it is not, the data itself is still compelling and I think the authors have demonstrated sufficiently that it cannot be explained trivially.

Reviewer #2 (Remarks to the Author):

In this new version of the manuscript, the main claim of the authors appears more clearly, making the manuscript more readable in general. The theoretical model has been refined, and its assumptions are more in line with the experimental observations. Several points have been in my view satisfactorily addressed, but a few others remain unclear and should be detailed prior to publication.

1) The amplitude of the change in the critical current is more appropriately discussed. The refined model brings an interesting explanation, based on the inversion of a single band, while 18 other bands remain trivial. The simulations confirm the correct magnitude of the observed transition. However, this model crucially depends on the fact that one band lies very close to the Fermi level. The authors say that this picture is «robust to small fluctuations in the density». Can the authors for example comment on the amplitude of density fluctuations from wire to wire or along the wire axis to assess the possibility that all measured wires verify this assumption?

2) The contribution of the single inverted mode amounts according to the authors to 74 nA, and is as such larger than the maximum critical current per channel $e\Delta/\hbar=50$ nA. How do the authors justify this discrepancy? Is this upper bound irrelevant in the limit $\mu \simeq \Delta$?

3) The origin of the finite resistance state should be better investigated. It is the sign of an important change in the junction's behavior. Its striking concurrence with the supercurrent enhancement points to a common origin. If this feature is expected in relevant theoretical models, it is worth using as an argument.

4) The authors have taken into account the exact geometry of the wires in the simulation of the magnetic field, which is a positive and important point. They however assume a perfect screening of the magnetic field, which is unlikely given the dimensions of the thin films with respect to the perpendicular and in-plane penetration depths in such thin films. How does this affect the results of

the magnetic field simulation?

To conclude, the revised manuscript presents the experimental observations and their interpretations more explicitly and more clearly. However, the interpretation seems to rely on a particular electronic configuration, whose reproducibility from one wire to another is too briefly discussed. Also, the importance of some other experimental facts is in my view underestimated, and requires deeper discussions. I believe those points should be clarified prior to publication.

Typo: at the end of page 11, the induced gap should read 200 μeV (and not meV)

Reviewer #3 (Remarks to the Author):

In this resubmission the authors have improved significantly the discussion on the possible theoretical explanation for the observed phenomenology. Most of the criticism raised in the previous round is addressed in a satisfactory manner and the proposed theoretical interpretation (based on the emergence of precursors of Majorana bound states) looks now like a plausible scenario. Of course, this is not a 'complete theory', but, considering the complexity of the system, it probably incorporates enough details to reasonably support the proposed scenario.

I have a few observations regarding the manuscript. The choice of colors in figures (2b) and (3a) is probably not the best (one can barely distinguish different sets of data). Also, I find the expression "band inversion" (associated with the topological quantum phase transition) a little bit confusing (since the sub-bands of the semiconductor wire do not change their order at the critical Zeeman energy). The more serious point concerns the angle dependence of the critical current enhancement (shown in Fig. 3). I do not understand why the case corresponding to the magnetic field parallel to the spin-orbit vector is not shown. After all, this should be a key piece of evidence for the proposed scenario, i.e. showing that one obtains a current enhancement at a finite Zeeman field when B is (or has a component) perpendicular to the spin-orbit vector, but there is no such enhancement when the two vectors are parallel. I believe that data corresponding to this field orientation has to be included and properly discussed. Finally, as pointed out by authors, the very low value of the critical Zeeman field (23mT) raises the most serious problem with the proposed interpretation. Clearly, there is considerable uncertainty in the parameters of the system. However, even considering a g -factor of 50, one still has to assume a very small induced gap (approximately $33 \mu\text{eV}$) to account for the observed value of the 'critical' Zeeman field. This, on the other hand, is probably not consistent with other estimates (e.g., the fits for the Josephson current). Solving this problem will probably require further studies, but I think that all its implications should be clearly stated (to avoid any possible confusion).

REVIEWERS' COMMENTS:

Reviewer #1 (Remarks to the Author):

I am satisfied with the new version and the answers provided to each of the reviewers. Note that the term "band inversion" is still used at the top of page 11, rather than the more appropriate "gap inversion".

Reviewer #2 (Remarks to the Author):

With this new version, the authors have made efforts to address the remaining issues I had regarding their manuscript. The paragraph added at the end of the theory section is in particular welcome. Some justifications remain debatable (items 1 and 3) in the absence of a more microscopic understanding of the system, and the topological origin of the observed phenomenon can not be completely established, as pointed out by Referee 1 and Referee 3.

Nevertheless, the authors have seriously addressed most of my points. They have convincingly ruled out the more trivial explanations, and a topological phase transition now appears as a more plausible candidate. Many more experiments are required to really understand the induced superconductivity in topological systems. This is true for the nanowire system of the present article, but goes beyond the scope of this article, and is also true for BiSe/BiTe/HgTe based topological insulators. In this context, and after this update, I can now recommend the publication of the present manuscript.

Reviewer #3 (Remarks to the Author):

The new material clarifies most of the issues raised in the previous round and provides reasonable explanations concerning some important limitations (e.g., why it is difficult to investigate a configuration involving a magnetic field parallel to the wire). Considering, as a whole, the experimental component of this work and the possible explanation based on a simple (but reasonable) model, I think that the manuscript should be considered for publication. My only recommendation would be to include a brief summary of the experimentally-relevant conditions that could generate deviations from the theoretical predictions based on the idealized model (e.g., finite size effects, electrostatic effects, etc.). It is important to clearly convey to the reader that some of the theoretical estimates (e.g., the critical field) are significantly different from the measured quantities and to provide possible causes for these (sometimes large) discrepancies.

REVIEWERS' COMMENTS:

Reviewer #1 (Remarks to the Author):

I am satisfied with the new version and the answers provided to each of the reviewers. Note that the term "band inversion" is still used at the top of page 11, rather than the more appropriate "gap inversion".

Reviewer #2 (Remarks to the Author):

With this new version, the authors have made efforts to address the remaining issues I had regarding their manuscript. The paragraph added at the end of the theory section is in particular welcome. Some justifications remain debatable (items 1 and 3) in the absence of a more microscopic understanding of the system, and the topological origin of the observed phenomenon can not be completely established, as pointed out by Referee 1 and Referee 3.

Nevertheless, the authors have seriously addressed most of my points. They have convincingly ruled out the more trivial explanations, and a topological phase transition now appears as a more plausible candidate. Many more experiments are required to really understand the induced superconductivity in topological systems. This is true for the nanowire system of the present article, but goes beyond the scope of this article, and is also true for BiSe/BiTe/HgTe based topological insulators. In this context, and after this update, I can now recommend the publication of the present manuscript.

Reviewer #3 (Remarks to the Author):

The new material clarifies most of the issues raised in the previous round and provides reasonable explanations concerning some important limitations (e.g., why it is difficult to investigate a configuration involving a magnetic field parallel to the wire). Considering, as a whole, the experimental component of this work and the possible explanation based on a simple (but reasonable) model, I think that the manuscript should be considered for publication. My only recommendation would be to include a brief summary of the experimentally-relevant conditions that could generate deviations from the theoretical predictions based on the idealized model (e.g., finite size effects, electrostatic effects, etc.). It is important to clearly convey to the reader that some of the theoretical estimates (e.g., the critical field) are significantly different from the measured quantities and to provide possible causes for these (sometimes large) discrepancies.

Reviewer #1 (Remarks to the Author):

REV 1 WROTE: The manuscript, by Paajaste et al., presents intriguing behaviour of critical currents in Al/InAs nanowire SNS junctions under an applied magnetic field. The experimental data and device characterization is laid out very clearly in figures 1-3 and the accompanying text. The critical current and hysteresis behaviour is indeed very similar to what has been reported in several previous papers by other groups. The new effect, a large jump in I_c that happens at a very similar field value in 3 different devices, certainly has not been reported on before, and does strongly compel one to ask what's happening. The authors reasonably argue that the effect is intrinsic, and not dependent on detailed device geometry. Further, the peculiar dependence on field orientation - with no (or very little) effect when the field is aligned with the spin-orbit vector (at least, reasonably assumed to be the SO direction) - does suggest a topological transition as a possible mechanism. If that were the case, the paper would indeed be a very interesting and worthy contribution.

After laying out all the experimental details, the authors turn to possible theoretical explanations - (1) a transition to a topological regime, (2) an exotic effect due to inhomogeneity of Zeeman splitting. This is the least convincing part of the paper, but a crucial one, and careful consideration raises a number of important questions which I believe the authors must address before this paper is further considered for publication.

OUR ANSWER: we agree with the reviewer, that our original explanation (1) of the topological transition was not detailed enough to explain the observed effect. We have refined it in the new version of the manuscript and, to avoid misinterpretations, we moved the second theoretical model (2) to the supplementary information. In the following we address all the specific questions of the Reviewer.

REV 1 WROTE: 1) A major criticism is that both theoretical models require very low transmittance to qualitatively explain the observations, whereas all experimental indications point to the opposite regime. The fact that supercurrents up to 150 nA are observed suggests relatively high interface transparency, and the semi-ballistic transport in the nanowire region means that overall transmission should be much higher than the order of 1% invoked to match the topological transition model. A ~1% transmittance would correspond to a tunnel junction, not an SNS junction, in my opinion. Infact, this inconsistency is probably large enough to comfortably rule out this particular explanation. Is there another mechanism by which the topological phase transition could increase I_c , but in the regime of high transmittance?

OUR ANSWER: The argument of the reviewer is indeed valid if the junction has a single occupied mode, which could not sustain such a large critical current at small transparency. However, a key to understand our critical current observations within our topological explanation is to realise that the charge density in the nanowire ($3 \times 10^{18} \text{ cm}^{-3}$) corresponds to a large number of occupied subbands. Non-topological critical currents in the order of 50 nA are consistent with ~ 20 low transparency ($T \approx 5\%$) spinful channels. We have performed detailed estimates of the expected number of occupied modes, using realistic parameters relevant to our samples, assuming a nanowire of hexagonal section, with an effective face-to-face distance of 38 nm. The resulting electronic structure yields 19 occupied (spinful) subbands, with the shallowest very close to the Fermi level $\mu_n \sim 0$, see Fig. C at the end of this letter. It is worth noting that this picture is robust over small fluctuations in the charge density or in the effective size of the nanowire (properties that are commonly sample specific) as (a) the geometry of the 2D confinement potential groups the subbands in narrow energy bands and (b) the compressibility around quasi-1D subband edges is small.

The $N=18$ filled spinful channels contribute to a background critical current $I_c \approx N T e\Delta/\hbar$, i.e. around 46 nA for a transmission $T \sim 5\%$ and an induced gap $\Delta \sim 200 \mu\text{eV}$. This background is largely independent of the magnetic field as long as the Al superconductivity is not destroyed. Upon the inversion of the low-lying

subband under the Zeeman field, the subband undergoes a topological transition and develops a Majorana mode at each side of the contact. This leads to an enhancement of the critical current from this subband. Within simple theory (assuming $\Delta \ll \mu_n$ and infinite wire length), the enhancement is given by $\Delta I_c \sim T^{1/2} e\Delta/2\hbar$. This, however, is a poor estimate if the assumptions of the theory are not fulfilled (in particular if the gap is not a small perturbation, and if transmission becomes strongly energy-, and thus Zeeman-dependent, as is the case of small $\Delta \sim \mu_n$). We have updated our simulations assuming $\Delta > \mu_n$ and a wire length $LS=500$ nm. The result is a contribution to the critical current by the lowest subband that goes from 13nA to 74nA upon inversion, for a normal transmission $T=5\%$ at $B=0$. Adding this contribution to the background from the remaining 18 modes at similar transmission yields good, almost quantitative agreement with the experimental results.

A key to understand the large ΔI_c enhancement obtained from our simulations upon the inversion of the shallowest subband, is to realize that the normal transmission T for this shallowest mode is strongly enhanced when it is Zeeman-polarized by B . Assuming $T \sim 5\%$ at zero field, for which the Fermi momentum along the wire is very small due to the small μ_n , we see an enhancement of the normal transparency T to $\sim 60\%$ at $B=1.5B_c$, which makes the jump in ΔI_c much more pronounced, from 13nA to 74nA. Roughly, the transmission depends monotonously on the (1D) momentum and therefore one can understand the increase of T in terms of the increase of the Fermi momentum along the wire as B is increased. This argument does not apply to other 18 subbands, as their Fermi momentum already starts out much larger at $B=0$. Each of these 18 deeper subbands contribute to I_c with a roughly B -independent ~ 45 nA background. The total I_c curves, summing background and the shallowest band, show rather good agreement with the experimental measurements (see figure A above). We emphasize that this calculation is fully microscopic, following the method of Ref. [21], and does not rely on the Andreev approximation. The results of simulation, including all parameters, are now provided in the updated version of the manuscript.

Corollary 1: the authors could easily estimate interface transparency using the so-called excess current (by measuring the I-V to higher biases) - this should be part of the basic device characterization.

OUR ANSWER: The excess current is a very useful information for ideal system in which the high-voltage normal state resistance is constant. Unfortunately this condition does not hold in our devices and the strong nonlinearities in the IV at high voltages do not allow to extract a reliable value for the excess current.

Corollary 2: it's a stretch to say that either model even qualitatively fits the observation; as field is increased I_c jumps fairly abruptly, then slowly decreases monotonically. Model 1 does not give an abrupt jump, in fact the major increase in I_c occurs well above B_{crit} (at a factor of 2 or 3 higher at least). For model 2, again there is not an abrupt jump, and it looks as if the I_c will go very quickly to zero after the peak (slope is incredibly steep).

OUR ANSWER: The original simulation for model 1 were indeed a merely qualitative description of the experiment, with chosen subband spacing and system parameters inadequate for our devices. The updated simulation, based on the correct charge density and geometry of our devices, is more accurate, and shows a much more abrupt I_c transition, happening closer to the critical value $B_c=\Delta$ for the topological transition of an infinite wire. The profile of I_c decrease after the transition. This is also in qualitative agreement with observations.

Other effects, for example stemming from electronic interactions are expected to enhance the abruptness of I_c jumps at subband inversions, as the total charge in the wire changes suddenly at that point. These effects, much harder to describe quantitatively, are neglected in our simulation, but are expected to increase transition sharpness even further. (Note however that, experimentally, the slope of the enhancement is always finite, i.e. it is not discontinuous, see e.g. Fig. 3 of the manuscript, or Fig. D at the end of this response).

Regarding model-2, we do not believe it can quantitatively explain our experimental observations, the scope of this model is to provide an alternative scenario in which there is an enhancement in B. To avoid confusion, we moved it in the supplementary materials where we clearly specify this point.

REV 1 WROTE: 2) The authors effectively rule out model 2 (inhomogeneous Zeeman field effect) because the inhomogeneity cannot be as large as the model would require, and they say as much in the abstract; however, it actually isn't made clear in the paper itself that they are ruling this out (which is confusing). I think it is useful and important that they show why this can be ruled out, but perhaps given that ultimately it is not a viable explanation, perhaps this could be included in Supplementary material rather than in the main paper.

OUR ANSWER: we agree with the reviewer and following his/her suggestion we moved model-2 in the supplementary.

REV 1 WROTE: 3) Another major criticism is that it's not clear whether the authors see this effect only in the central junction or not. The picture of Fig 4a suggests that the topological model can only explain the data with this rather ad hoc assumption that the outer junctions behave very differently from the central junction. I don't see any justification from the device geometry to support this. Do the authors have transport data to back this up? Presumably they can also measure the I-V's of the outer junctions, and report whether the same overall SNS behaviour is observed, and indeed whether this phenomenon (jump in I_c) is observed. If those junctions show similar behaviour, it would appear to rule out the picture of Fig 4a. Certainly, it would go a long way to clarifying the paper if the authors showed the I-V characteristics of all three junctions on one nanowire for comparison. It is mentioned that 10 junctions were fabricated, but data from only 3 are shown...can the authors tell us how many of the junctions worked and also showed this same critical current behaviour?

OUR ANSWER: We thank the referee for raising this point, which is a powerful argument that could, at first sight, be able to discern between the Majorana and alternative scenarios. In our experiments, critical currents always exhibit, to some extent, an increase at a certain magnetic field, regardless of which (adjacent) pair of contacts is used. The most robust jumps, however, are observed in the contacts at the middle of the wire.

Given that at the outermost contacts, one side of the junction is about 100 nm (and the other side is much longer), it might appear implausible that Majoranas can still contribute a sizeable supercurrent there, as the hybridization of spatially overlapping Majoranas should destroy the effect. However, a quantitative simulation (see Figure B at the end of this letter) shows that this is not the case close to subband depletion $\mu_n \sim 0$ (which is the scenario explored here). Indeed, the I_c enhancement is rather generic down to $L \sim 100$ nm (of the order of the spin-orbit length). The enhancement in our simulations becomes smaller gradually, as the junction is moved towards the edge of the wire, and the transition B is increased.

The reason for the existence of these remnants of the topological transition and Majoranas is that, while outer and inner Majoranas on the short side of the junction overlap spatially, their energy splitting away from zero remains very small close to the transition at $B=B_c$, effectively mimicking a longer wire to some extent. As explained in the updated manuscript, this is due to the smallness of μ_n , as the Majorana splitting is known to scale with the Fermi wave number k_F , see Refs. [38,39]. (This suppressed splitting at $\mu_n \sim 0$ is also the reason why the Majorana oscillations obtained theoretically are unusually small close to B_c in such case, even for short nanowires, and grow only as B is increased further, increasing k_F .)

This theoretical picture remains consistent with our experimental observations, which thus strengthens our topological interpretation. From the experimental point of view, the only device that allows a concrete comparison between edge and middle junctions is reported in Fig. D at the end of this reply. In this device

all the junctions have similar length ($L \sim 110$ nm) and we can notice a more pronounced enhancement in the middle one. Due to the loss of statistics over many NWs from this measurement we can only extract a general trend so we preferred not to discuss it in the main text.

REV 1 WROTE: 4) Gating of the nanowire is not discussed, even though it is fabricated on n^{++} Si and therefore has a back gate. I assume this is because the nanowires are heavily doped and so the gate is ineffective. Is this correct? If the devices can be gated at all, it would have been very interesting to see the dependence of this effect on changing the chemical potential. For example, the even-odd effect with respect to number of occupied subbands could be tested.

OUR ANSWER: The suggestion of the reviewer is correct, unfortunately the nanowires are so heavily doped that we were not able to significantly effect the electron density by back gating even after the application of few volts. We preferred not to include this minor detail in the main text.

REV 1 WROTE: 5) The authors should measure the normal state conductance (e.g. at a temperature just above the SC transition temperature) versus field to rule out some other magnetoconductance effect unrelated to superconductivity. I agree it is unlikely, but to be sure this should be checked.

OUR ANSWER: This is also an important point and therefore we have studied the normal state conductance thoroughly. Above the critical temperature of the Al-leads (2K), no change in the magnetoresistance is visible in the full range of magnetic fields explored. Thus, we can safely rule out the hypothesis of an enhanced critical current due to the magnetoresitences effects. To clarify this point we included this discussion in the amended version of the manuscript.

REV 1 WROTE: To summarize: I believe the authors have observed an intriguing effect, and have done a fairly convincing job to show that it is likely an intrinsic, non-trivial effect. However I believe there are simple things they can do that will go a long way to clarifying what's really happening: estimating contact transparency from I-V data, reporting on all junctions in a single device, measuring gate dependence (if this is possible), and checking normal state conductance versus field. As it stands, the models proposed appear to be unsatisfactory for explaining, even qualitatively, the observed behaviour. For these reasons, I recommend the manuscript not be published in its current state.

OUR ANSWER: we thanks the reviewer for the useful comments that help us improved the manuscript and the consistency of the theoretical model.

Reviewer #2 (Remarks to the Author):

REV 2 WROTE: This paper reports measurements of critical currents in Josephson junctions made of InAs nanowires as a weak link. The authors identify a strong enhancement of the critical current when a magnetic field is applied and reaches a critical value, that depends on the magnetic field orientation. They examine the dependence of the effect on several parameters (magnetic field direction, geometrical dimensions, temperature) and assess the relevance of various scenarii. They conclude that « the observed I_c enhancement is compatible with a magnetic field induced topological transition of the junction ».

In general, the measurements are clean. The study of several relevant parameters is thoroughly carried out and several devices have been studied, giving a systematic character to the observations. In consequence, the experimental evidence for the sudden enhancement of the critical current is convincing.

The authors also conduct a study of different theoretical explanations but however fail in giving very strong arguments in favor of the occurrence of a topological transition in their system. The following points in that arose during my reading of the manuscript. Points 1), 2) and 3) are of particular importance.

OUR ANSWER: we thanks for all the criticism pointed out by the reviewer. We substantially refined our theoretical model accordingly, taking in to account the full electronic structure of the nanowire. We are now able to describe the enhancement fo the Josephson current with a quasi quantitative topological model. In the following we carefully address all the points raised by the reviewer.

REV. 2 WROTE: 1) Providing microscopic information on the device is necessary to justify the relevance of the topological transition of Ref.20. For example, this theory applies mainly to a low transmission regime while the authors do not show particular indications that this regime is applicable here. Besides, though the authors qualify the effect of « colossal », the amplitude of the critical current enhancement is not used as an argument to substantiate any of the presented scenarii. Having estimated the induced gap and the electron density, I would imagine that the authors could evaluate the number of modes, their average transmission, and comment on the amplitude of the change in I_c . It is a rather important point to address, in order to justify how the appearance of a limited number of modes results in a massive doubling of the critical current. Could the authors comment on that?

OUR ANSWER: This is an important point that was also brought up by another reviewer. The original manuscript lacked a proper analysis of the expected electronic structure of the nanowire, given its charge density ($n=3 \cdot 10^{18} \text{ cm}^{-3}$), and a corresponding simulation of the expected critical current versus field expected for the device parameters. In the present version of the manuscript we have corrected this problem with an updated and more detailed simulation. The estimated number of filled spinful subbands, using a nanowire with hexagonal section and a 38 nm face-to-face distance, is 19, see Fig. C at the end of this letter. Given the critical field at zero magnetic field $I_c \sim N T e \Delta / \hbar \sim 50 \text{ nA}$, with induced gap $\Delta \sim 200 \mu\text{eV}$, we estimate a low average transmission per channel of $T=5\%$.

When applying a Zeeman field, the lowest lying subband, that according to our calculations should lie very close to the Fermi energy, $\mu_n \sim \Delta$, becomes inverted. This is expected to give rise to Majorana bound states at the contact, which support an enhanced critical current, usually estimated as $\Delta I_c = T^{1/2} e \Delta / 2 \hbar$. This estimate, however, is based on an infinitely long wire in the Andreev approximation $\mu \gg \Delta$ and a constant transmission T , and severely underestimates the actual I_c enhancement for small $\mu \sim \Delta$. An actual simulation, with all parameters adjusted to our experimental devices, yields an increase in the I_c contribution of the inverted subband that goes from 13nA to around 74nA after the topological transition.

Apart from the topological enhancement $T \rightarrow T^{1/2}$, one further reason for the I_c enhancement is that the normal transmission T for this shallowest mode grows as it is Zeeman-polarized by B . Assuming $T \sim 5\%$ at zero field, for which the Fermi momentum along the wire is very small due to the small μ_n , we see an enhancement to $T \sim 60\%$ at $B = 1.5B_c$, which makes the jump in I_c even more pronounced. The increase in T is due to the increase in the Fermi momentum along the wire as B is increased, and the roughly monotonous dependence of transmission with said momentum in 1D. The same does not apply to other 18 subbands, as their Fermi momentum already starts out much larger at $B = 0$. Each of these 18 deeper subbands contribute to I_c with a roughly B -independent $18 \cdot 5\% \cdot e\Delta/\hbar \sim 45 \text{ nA}$ background. The total I_c curves, summing background and the shallowest band, show good agreement with the experimental measurements (see figure A at the end of this letter). Full details on the new model, simulation and parameters are now given in the updated manuscript.

REV. 2 WROTE: 2) The properties of superconducting thin films such as the ones used in this study can be particularly complicated in a magnetic field. In particular, the film has to overcome the non-negligible height of the nanowire, so that a «perpendicular» magnetic field is never totally normal to the superconducting surface. Do the simulations of Fig.3b take this effect into account? Have the authors established experimentally the behavior of their films via studies of its resistance or of its diamagnetic properties?

OUR ANSWER: The simulations reported in Fig.3b account for the particular shape of the leads overcoming the nanowire (shown in the sketch of Fig1b). This shape cannot be appreciated from the top view of Fig.3b so we describe this point better in the legend of the figure "Vertical gray regions indicate the superconducting electrodes overcoming the light-green horizontal NW." Due to the complexity of this geometry, the simulations consider ideal screening of B . We did not specifically measure the resistance and the diamagnetic properties of our specific leads, due to the difficulties in setting a 4w measurement on a nanoscopic open leads, for this purpose a different geometry for the device should be used.

REV. 2 WROTE: 3) As noticed by the authors, the I-V characteristics reveal the appearance of a residual dissipative behavior in the supercurrent (with a resistance of approximately 60 Ω). No possible reasons is given for this surprising feature in the article, which in fact even questions the designation « supercurrent ». Can the authors speculate on origins for the observed phenomenon? At this stage, it is difficult for me to exclude quenching of superconductivity in the narrow Al superconducting leads, as studied for example in Verduyssen et al., PRB 85, 224503. Especially Figs. 6 and 7 are quite similar to Fig. 2b of this paper. Together with 2), one could at first sight argue that the supercurrent in fact collapses (no enhancement) and the system then gets into an intermediate state of resistance 60 Ω . Can the authors exclude this behavior (from measurements of the leads resistance for instance)?

OUR ANSWER: The size of the narrow Al wires ($w = 100\text{-}200 \text{ nm}$, $t = 50\text{-}90 \text{ nm}$) studied by Verduyssen et al. (PRB 85, 224503) is very comparable to our Al leads ($w = 150 \text{ nm}$, $t = 100 \text{ nm}$). While at this size they measure a critical current higher than $10 \text{ }\mu\text{A}$ for all the Al wires (fig6 and 7), this is enough to rule out a possible quenching of the superconductivity in our Al leads in which the maximum critical current measured ($\sim 150 \text{ nA}$), driven by the NW Josephson junction, is far below these values. The observation of a small critical current in the NW is sufficient to exclude this behaviour.

We could speculate on the origin of the dissipative behavior considering the closing of the gap at the topological transition. There the system can have a dissipative component (see. e.g. Fig. 4 in New J. Phys., **15**, 75019, (2013)).

REV. 2 WROTE: 4) In a related note, the authors have previously demonstrated the possibility to use Pb contacts to realize Josephson junctions on InAs nanowires. But as far as I understand, these devices do not exhibit the same enhancement of the critical current when a magnetic field is applied, but a more

conventional Fraunhofer pattern. Could the authors comment on this fact, and on the advantage of using Al?

OUR ANSWER : The point raised by the reviewer is very interesting. Indeed in the experiment mentioned and reported in reference [J. Paajaste, Nano Lett., vol. 15, 3, 1803, (2015)] we measured a shoulder in I_c at $B \sim 130$ mT. This is consistent with the “topological” switching field expected for Pb-based devices obtained from the renormalization of the measured switching field for the aluminum-based junctions with the proper pairing potential ($23 \text{ mT} * \Delta(\text{Al}) / \Delta(\text{Pb}) \sim 138 \text{ mT}$). Moreover the weaker enhancement observed in this experiment could be justified by a different transparency between the Pb and Nanowire interface.

It is difficult for the Pb experiments to understand whether the origin of the anomalous shape of I_c is due to Fraunhofer interference or a topological transition. For this one would need a characterization of the critical current for different angles of the magnetic field that has not been done for the Pb-based devices.

For all these reasons, the Al-based devices are more suitable for this specific topological study even if a comparison with different materials could be useful to confirm the relationship between the critical field of the topological transition and the pairing potential. We added a comment in the main text to underline this analogy.

REV. 2 WROTE: 5) It has been proposed (in particular by some authors of the present paper) to examine the threshold for finite voltage features such as the excess current or multiple Andreev reflections. This onset should indeed be observed at Δ/e in the presence of Majorana Bound States (and not $2\Delta/e$ in the conventional case). Have the authors tried to establish the presence of an excess current and to observe changes in its onset around the observed transition ?

OUR ANSWER: Excess current and Multiple Andreev reflection (MAR) measurements are indeed a potentially valuable probe into the system. We have performed preliminary measurements of both excess current and MAR. However, an accurate analysis of these phenomena would require an extensive experimental study not possible for us at this stage. As written in the answer to Ref. 1, our preliminary analysis is difficult to evaluate due to the nonlinearities observed in the normal state resistance at high voltages, while MAR show complex behavior at the I_c transition which would require considerably more modelling work.

REV. 2 WROTE: To conclude, the paper is well written and relies on clean experimental data. A clear demonstration of a topological transition in InAs nanowires would in my opinion qualify for a publication in Nature Communications, but such a claim is insufficiently justified in the current state of the manuscript. Crucial information is still needed to support the authors' interpretation. The authors could possibly make use of the previously raised issues to further confirm (or infirm) the hypothesis of a topological transition.

OUR ANSWER: We believe that with the new refined version of the theoretical model we clarified the important role played by the topological transition in our InAs nanowires. We thank the reviewer for pointing out the analogy and differences between the Al- and Pb-based devices supporting the topological hypothesis even further.

Reviewer #3 (Remarks to the Author):

REV. 3 WROTE: In this manuscript the authors report the observation of a large (up to about 100%) enhancement of the critical supercurrent at finite magnetic fields in Josephson junctions based on semiconductor nanowires proximity coupled to s-wave superconductors. The system under consideration has all the key ingredients that are required for the realization of topological superconductivity (and the corresponding Majorana bound states) in solid state hybrid structures: strong spin-orbit coupling, (proximity-induced) superconductivity, and magnetic field (to break time-reversal symmetry). From this perspective, the work described here fits well into the larger ongoing effort to understand the physics of hybrid structures that may support nontrivial topological phases and exotic quasiparticle excitations. Moreover, in my opinion, the main finding of this work is very interesting and rather puzzling. On the other hand, I find the possible explanations of the observed phenomenon to be unconvincing and potentially misleading.

I recognize that the system is complex and that modeling it theoretically represents a difficult task. However, the scenario suggested by the authors as being the most likely (enhancement generated by the system undergoing a magnetic field-induced topological phase transition) is not very well supported by the experimental evidence. Here are some of the aspects that I find troubling.

OUR ANSWER: we agree with the criticism of the Reviewer on the oversimplified version of the topological model. In the following we reply to all the point raised by the reviewer showing how the new refined version of the theoretical model can almost quantitatively describe all the experimental observations.

REV. 3 WROTE: 1)Dependence on the orientation of the magnetic field. Most of the signatures consistent with the presence of Majorana bound states (MBSs) that were observed so far were obtained with a magnetic field oriented parallel to the wire. I do not see a convincing reason for having MBSs for a field oriented perpendicular to the substrate and not having them for a field parallel to the wire. The magnetic focusing effect may be responsible for some quantitative differences, but it cannot change the physics qualitatively. On the other hand, I can see a strong reason for NOT having MBSs in the perpendicular geometry. Basically, the Zeeman field under the Ti/Al superconductor (where induced topological superconductivity is supposed to emerge) is very small and it would be extremely surprising to find that it satisfies the topological condition. Also, assuming that the structure from Fig. 1a can be (qualitatively) modeled by the structure shown in Fig. 4a is a stretch. Why would the uncovered middle segment be any different from the other two segments? If they are not, there should be four pairs of strongly overlapping MBSs (instead of two).

OUR ANSWER: This is a crucial question, and we realise the original manuscript didn't clarify this issue sufficiently.

Regarding the first part (why parallel field don't work), it is a matter of field screening by the overhanging superconducting fingers. As shown by our COMSOL simulations, calculated exactly for the shape of our superconducting leads completely overcoming the NW (as specified in the new legend of fig.3), a parallel field is almost completely deflected away from the nanowire by the fingers. This is consistent with the much weaker field dependence observed in this configuration that is very different from the scenario characterizing similar experiment in which, to avoid field repulsion, the nanowire is only partially covered by the Al lead (e.g. Muorik, Scinece, 336, 1003, (2012)).

Regarding the second part, it is indeed true that the same screening mechanism applied to perpendicular fields deflects the field away from the covered parts of the wire, but now concentrates it within the space separating the fingers. Hence, for perpendicular fields, carriers in the wire experience a spatial alternation between pairing and Zeeman coupling. Covered parts have induced pairing (all with the same phase on a

given side of the junction) and negligible Zeeman. Uncovered parts have focused Zeeman and no pairing. The key why this works for Majorana formation is that carriers from the shallowest mode (which is the one undergoing an inversion) have a wavelength that ranges from $\sim 200\text{nm}$ to $\sim 95\text{ nm}$ as the field goes from zero to $1.5 B_c$, so carriers from this mode remain highly delocalised along the wire. In this situation, we have checked numerically that the relevant Zeeman and pairing they feel is essentially a spatial average of both, which makes sense as any spatial modulation in the Hamiltonian is washed out on the scale of carrier wavelength. The exception is a jump in the superconducting phase, which occurs at the current-biased junction depending on the injected supercurrent. In this case the phase jump controls Majorana hybridization regardless on how abrupt it is. We have included this discussion in the revised manuscript.

REV. 3 WROTE: 2)Short length wires. The superconducting segments of the wire are very short; one cannot talk about a topological phase transition in these conditions. In the best case scenario one can have some strongly overlapping bound states that can be interpreted as "precursors" of the MBSs, but the emergence of these states (e.g., the "critical" field associated with their occurrence and the field range over which they are present) depend strongly on the details of the system.

OUR ANSWER: We have overhauled our model and simulation to achieve a quantitative description of our devices in the proposed topological scenario, including both the wire density, wire geometry and radius, and the wire length $\sim 500\text{ nm}$ at each side of the junction, that as pointed out by the Reviewer, is crucial in the context of a topological transition. Our updated scenario involves 18 occupied subbands, and a shallow subband with small Fermi energy $\mu \sim \Delta$ that becomes inverted by the Zeeman field. In the limit of small μ we have shown that while Majoranas at a given side of the junction indeed have substantial spatial overlap, they remain almost orthogonal if μ is small, and hence behave almost as perfect Majoranas in an infinite wire. The spectrum in Fig. A(right), see end of this letter, shows this effect. We have confirmed that this behavior is generic for small μ , and can be seen in many works in the literature(e.g. Klinovaja, PRB **86**, 085408 (2012) and S. Das Sarma, PRB **86**, 220506(R) (2012)). Indeed, the Majorana oscillations in finite-length nanowire always start with very small amplitude at $B \sim B_c$ if μ is small, regardless of system details. This is exactly the situation in our samples.

REV. 3 WROTE: 3)Other problems. The estimated induced gap is not consistent with the observed critical field. There is a factor of 8 between the two quantities; even considering a focusing factor of 2 the difference remains significant.

OUR ANSWER: The Referee's point is correct if we assume a single occupied subband in the nanowire. Given its charge density $n = 3 \cdot 10^{18}\text{ cm}^{-3}$ and diameter $\sim 40\text{ nm}$, this is not the appropriate regime. In our updated theory description we take this into account, and find that there are actually around 19 spinful occupied subbands. Only the topmost subband, with $\mu \sim \Delta$, becomes inverted by the magnetic field. The induced gap $\Delta \sim 200\mu\text{eV}$ is then consistent the critical current at zero field $I_c \sim 50\text{nA}$ for a transmission of $T = 5\%$. The updated model also predicts an twofold increase of this I_c under the inversion of a single band. The explanation is the following

When applying a Zeeman field, the lowest lying subband, that according to our calculations should lie very close to the Fermi energy, $\mu \sim \Delta$, becomes inverted. This is expected to give rise to Majorana bound states at the contact, which support an enhanced critical current, usually estimated as $\Delta I_c = T^{1/2} e \Delta / 2\hbar$. However, this estimate, based on an infinitely long wire in the Andreev approximation $\mu \gg \Delta$ and a constant transmission T , severely underestimates the actual I_c enhancement for small $\mu \sim \Delta$. An actual simulation, with all parameters adjusted to the experimental devices, yields an increase in the I_c contribution from 13nA to around 74nA from the lowest subband inversion alone.

Apart from the topological enhancement $T \rightarrow T^{1/2}$, one further reason for the I_c enhancement is that the normal transmission T for this shallowest mode is grows as it is Zeeman -polarized by B . Assuming $T \sim 5\%$ at

zero field, for which the Fermi momentum along the wire is very small due to the small μ_n , we see an enhancement to $T \sim 60\%$ at $B = 1.5B_c$, which makes the jump in I_c even more pronounced. The increase in T is due to the increase in the Fermi momentum along the wire as B is increased, and the roughly monotonous dependence of transmission with said momentum in 1D. The same does not apply to other 18 subbands, as their Fermi momentum already starts out much larger at $B=0$. Each of these 18 deeper subbands contribute to I_c with a roughly B -independent $18 \times 5\% \times e\Delta/\hbar \sim 45\text{nA}$ background. The total I_c curves, summing background and the shallowest band, show good agreement with the experimental measurements (see figure A at the end of this letter). Full details on the new model, simulation and parameters are now given in the updated manuscript.

REV. 3 WROTE: Moreover, applying the factor of 2 uniformly is wrong. If a state extends throughout the whole L_s region, the effective Zeeman field that it "feels" is closer to the bare field (not the focused one). If, on the other hand, we consider a state localized right outside the superconductor, then it has nothing to do with the topological phase transition (that characterizes the superconducting segment of the wire, which has a much lower Zeeman field).

OUR ANSWER : The reviewer is right, and we have corrected the manuscript to reflect this. The carriers from the shallowest subband feel a spatial average of the Zeeman field (since their wavelength, ranging from 200nm to 95 nm, is larger than the spacing between superconducting fingers). The Zeeman field is essentially zero under the contacts, and essentially doubled between the contacts, which gives the bare value for the spatial average.

REV. 3 WROTE: In conclusion, the manuscript describes an interesting phenomenon, but provides no convincing explanation for the observed physics. In addition, the study could have been more thorough; a straightforward test that could provide further support for the topological transition scenario (or eliminate it) would have involved longer superconducting regions and (possibly) partially covered wires (with a magnetic field oriented along the wire). In the absence of these tests, the work presented in the manuscript poses some interesting questions, but does not clarify the outstanding problems already existing in this field.

OUR ANSWER: We believe this conclusion is a consequence of the merely qualitative modelling effort performed in the original manuscript. We trust the Reviewer will find our updated theoretical analysis much more convincing, as it naturally reproduces the observed I_c phenomenology at an almost quantitative level.

FIGURES, COMMON TO ALL REPLIES

Fig A: (Left) Critical current for a nanowire with 19 occupied modes (hexagonal section, 38 nm face to face distance, density $n=3 \cdot 10^{18} \text{ cm}^{-3}$). The shallowest mode is close to depletion, and the average normal transmission is $\sim 5\%$. The increase in critical current I_c comes from the inversion of the almost depleted mode, and the emergence of Majoranas at the junction. (Right) The Andreev spectrum at $B=1.5 B_c$ (red) and $B=0$ (dotted black). Each side of the junction is 500 nm long.

Fig B: Same as figure A but with an asymmetric junction. The left side is 900 nm long, and the right side is 100 nm long. The spectrum in red corresponds to $B=2.5 B_c$.

Fig C: Energies for subband edges, computed for a hexagonal nanowire of 38nm face-to-face distance. Red lines mark the Fermi energy for density $n=3 \cdot 10^{18} \text{ cm}^{-3}$ (19 occupied spinful subbands). Note that 1 subband edge lies very close to the Fermi energy.

Fig D: (left) SEM image of one of the NW with three Josephson junctions of similar length. (right) Critical current vs out-of-plane magnetic field showing a more pronounced enhancement for the middle junction.

Reviewer #1 (Remarks to the Author):

REV1 WROTE: I've carefully reviewed the new material, and find the replies to my queries are sufficient and the revised manuscript much improved. The more complete characterization of the devices in terms of number of occupied subbands together with a more convincing theoretical model make the topological explanation much more plausible. In particular, the explanation that I_c scales with the square root of the transmission for the zero modes, and that transmission is of order 5%, resolves one of main problems I had with the paper. The only worrisome thing is the remaining discrepancy between the predicted and observed critical fields. The authors note that this could be explained by a combination of factors, such as increased g-factor due to confinement and non-ideal proximity gap. Do they also mean to include the field focusing effect? That would give a factor of two, which would mean the other factors only need give a factor ~ 10 , which is more plausible than a factor 20. Despite this issue, I think overall their picture is a good candidate to explain the observations. I can't say with 100% certainty that it is the correct explanation, but this new version lays out a much more compelling argument than the first version. I would now support publication. I think it likely that the topological explanation is correct. However, even in the (unlikely) event that it is not, the data itself is still compelling and I think the authors have demonstrated sufficiently that it cannot be explained trivially.

OUR ANSWER: We appreciate the positive assessment of the reviewer. After the last round we performed a careful analysis of the effect of focusing (which we expect is indeed real in our samples) on the electronic structure of the nanowires. We concluded that in the regime relevant for the experiment, the shallowest subband is effectively subject to a spatially averaged g-factor, so that focusing should have little effect on its g-factor. There are, however, other potentially powerful effects that could enhance the g-factor of these samples. Interactions with the substrate, with the superconductor and electronic reconstruction on the nanowire surface are some of these, but a more precise analysis is unfortunately rather far out of the scope of the present work.

Reviewer #2 (Remarks to the Author):

REV2 WROTE In this new version of the manuscript, the main claim of the authors appears more clearly, making the manuscript more readable in general. The theoretical model has been refined, and its assumptions are more in line with the experimental observations. Several points have been in my view satisfactorily addressed, but a few others remain unclear and should be detailed prior to publication.

1-The amplitude of the change in the critical current is more appropriately discussed. The refined model brings an interesting explanation, based on the inversion of a single band, while 18 other bands remain trivial. The simulations confirm the correct magnitude of the observed transition. However, this model crucially depends on the fact that one band lies very close to the Fermi level. The authors say that this picture is «robust to small fluctuations in the density ». Can the authors for example comment on the amplitude of density fluctuations from wire to wire or along the wire axis to assess the possibility that all measured wires verify this assumption?

OUR ANSWER: The reviewer stressed a very interesting and crucial point of our theoretical model. Despite the accurate evaluation of the energy levels in the NW obtained simulating the hexagonal confinement small differences ($\sim 10\%$) in the diameter of the nanowires are commonly observed. In the ideal scenario this small change can move the Fermi level far ($\sim \text{meV}$) from the last occupied subband making the topological condition impossible to be achieved with a small magnetic field of $\sim 23 \text{ mT}$.

As suggested by the reviewer the real experimental scenario is much more complex respect to the standard 1D confinement used in our estimation. Specifically band banding is expected at the interface between the NW and the Al leads moving the energy levels along the wire and then favoring the crossing with the Fermi level. For a better understanding of this effect, a more accurate and quantitative description of the electronic band structure along the NW is necessary, this very demanding calculation is far from the scope of the standard topological model presented to support the experimental data.

2-The contribution of the single inverted mode amounts according to the authors to 74 nA, and is as such larger than the maximum critical current per channel $e\Delta/\hbar=50 \text{ nA}$. How do the authors justify this discrepancy? Is this upper bound irrelevant in the limit $\mu \simeq \Delta$?

OUR ANSWER: We thank the reviewer for bringing this up. The bound is indeed relevant, and should approximately hold, even away from the Andreev approximation. We have carefully revised our numerical results to find out why they do not satisfy this basic constraint on the maximum supercurrent $e\Delta/\hbar$ through a spinful channel. We have found a mistake in our numerical code (wrong numerical prefactor) that was regrettably leading to an overestimation of the critical current, and have now corrected it. The general conclusions of the theory section, however, have not changed. The revised numerical results have required a slight adjustment of the estimated system parameters, in particular the induced gap Δ and spin-orbit strength α_{SO} , which are now assumed stronger by about 25%, and the average normal transmission T_{N} that is slightly reduced to recover the experimental results. The maximum critical current possible for a single *spinless*

channel is now $e\Delta/2\hbar=30$ nA, and the actual maximum in the particular simulation of Fig. 4 is 10 nA. This is enough to model the measurements on longer junctions ($L=114$ nm, orange points in experimental Figure 2c). The shorter junctions require more care, however, as their I_c enhancement exceeds the single-mode supercurrent limit, and points to the possibility that more than one shallow subband could be contributing to the jump in critical current. This is a possibility we had discussed internally, but decided to leave out of the draft to simplify the theoretical picture. Indeed, as can be seen in the subband spectrum of Fig. 2 (Supplementary Information), subbands cluster in pairs. A slight enhancement of the charge density, still compatible with our measurements, would push the Fermi energy close to one of these pairs, which would give a doubled I_c enhancement, to a maximum of $\sim 60-70$ nA, closely matching our highest values of supercurrent enhancement. (The coexistence of two topological subbands in a single junction is possible within the BDI symmetry class, wherein the nanowire thickness is quite smaller than the spin-orbit length, as is the case here.) Given the complexity of this scenario, however, we have chosen to focus on the single topological subband and the longer junctions for the numerics, while clearly mentioning the issue pointed out by the reviewer in relation to the shorter junctions.

3-The origin of the finite resistance state should be better investigated. It is the sign of an important change in the junction's behavior. Its striking concurrence with the supercurrent enhancement points to a common origin. If this feature is expected in relevant theoretical models, it is worth using as an argument.

OUR ANSWER: We agree that this is a very significant clue as to the nature of the transition. The simplest explanation for a dissipative component in the supercurrent is the development of a soft gap in the junction that develops simultaneously with the supercurrent enhancement. The development of gap softness is consistent with a topological transition, and in fact we think it strongly supports the proposed explanation. The reason is that after a Zeeman-induced gap inversion, the topological gap is no longer s-wave, but a topological p-wave, supported solely by the spin-orbit coupling, and not protected against disorder by the Anderson theorem. It is actually a much discussed experimental observation from the first Majorana experiments [e.g. Mourik et al. Science 2012], that soft gaps are invariably observed after the assumed topological transition in Rashba nanowires. We thus expect that the dissipative current component would be suppressed by enhancing the spin-orbit coupling or by decreasing disorder in the junction. We have currently no means to perform such investigation experimentally, but have included a discussion of the above at the end of the theory section in the updated manuscript.

4-The authors have taken into account the exact geometry of the wires in the simulation of the magnetic field, which is a positive and important point. They however assume a perfect screening of the magnetic field, which is unlikely given the dimensions of the thin films with respect to the perpendicular and in-plane penetration depths in such thin films. How does this affect the results of the magnetic field simulation?

OUR ANSWER: Indeed, the finite element simulation assumes perfect screening, an approximation that should fail to some extent given the thickness of the superconducting electrode. The main point drawn from these simulations is that in plane fields along the wire should be better screened away from the nanowire electrons than out of plane fields. This should be always true, even if the

assumption of perfect electrode screening is not entirely fulfilled. The reason is that the field penetration length, while probably comparable to the thickness of a single electrode, is smaller than the width of the whole set of fingers in each sample. In this regard, the main result of that simulation, i.e. a very different Zeeman coupling in the wire for in-plane and out-of-plane fields, is expected to hold, which is validated by the experimentally observed response in transport.

REV2 WROTE: To conclude, the revised manuscript presents the experimental observations and their interpretations more explicitly and more clearly. However, the interpretation seems to rely on a particular electronic configuration, whose reproducibility from one wire to another is too briefly discussed. Also, the importance of some other experimental facts is in my view underestimated, and requires deeper discussions. I believe those points should be clarified prior to publication.

Typo: at the end of page 11, the induced gap should read 200 μeV (and not meV)

OUR ANSWER: We thank the reviewer for the positive comments on the revised manuscript and we believe that in this new version we could clarify also the few missing points for a reasonable interpretation of the experimental data. Of course, our model, as recognized by reviewer #3, does not pretend to be a 'complete theory', but, considering the complexity of the system, it probably incorporates enough details to reasonably support the proposed scenario.

Reviewer #3 (Remarks to the Author):

REV3 WORTE: In this resubmission the authors have improved significantly the discussion on the possible theoretical explanation for the observed phenomenology. Most of the criticism raised in the previous round is addressed in a satisfactory manner and the proposed theoretical interpretation (based on the emergence of precursors of Majorana bound states) looks now like a plausible scenario. Of course, this is not a 'complete theory', but, considering the complexity of the system, it probably incorporates enough details to reasonably support the proposed scenario.

I have a few observations regarding the manuscript.

1-The choice of colors in figures (2b) and (3a) is probably not the best (one can barely distinguish different sets of data).

OUR ANSWER:

We believe that 2b is clear enough as each curve is addressed not by the color but by its own value specified in the curve itself. We will consider to modify the colors of Fig 3a on the final proofs of the manuscript.

2-Also, I find the expression "band inversion" (associated with the topological quantum phase transition) a little bit confusing (since the sub-bands of the semiconductor wire do not change their order at the critical Zeeman energy).

OUR ANSWER: We agree with the reviewer and we have changed the expression to the more common and precise "gap inversion", following the terminology of e.g. *Nature Physics* **12**, 618–621 (2016) or *Annual Review of Condensed Matter Physics* **4**, 113-136 (2013)

3-The more serious point concerns the angle dependence of the critical current enhancement (shown in Fig. 3). I do not understand why the case corresponding to the magnetic field parallel to the spin-orbit vector is not shown. After all, this should be a key piece of evidence for the proposed scenario, i.e. showing that one obtains a current enhancement at a finite Zeeman field when B is (or has a component) perpendicular to the spin-orbit vector, but there is no such enhancement when the two vectors are parallel. I believe that data corresponding to this field orientation has to be included and properly discussed.

OUR ANSWER: The point raised by the reviewer is correct, unfortunately for the requested characterization a vectorial magnet is necessary to have the fine control over the orientation of the magnetic field. Our measurements are performed with a single-axis magnet and the different orientations of the field are performed with the mechanical tilting of the sample that does not allow for a precise alignment of the in-plane magnetic field respect to the spin orbit vector.

4-Finally, as pointed out by authors, the very low value of the critical Zeeman field (23mT) raises the most serious problem with the proposed interpretation. Clearly, there is considerable uncertainty in the parameters of the system. However, even considering a g-factor of 50, one

still has to assume a very small induced gap (approximately $33 \mu\text{eV}$) to account for the observed value of the 'critical' Zeeman field. This, on the other hand, is probably not consistent with other estimates (e.g., the fits for the Josephson current). Solving this problem will probably require further studies, but I think that all its implications should be clearly stated (to avoid any possible confusion).

OUR ANSWER: We agree that the above is an issue with our simple theory. The non-interacting, BCS mean-field model for the topological transition used in our theory section indeed predicts quite a larger value for the critical field than observed. There are, however, several assumptions in this simple standard model that may not be applicable in real samples. We are still in the early days of a quantitative understanding of topological superconducting systems, and it is likely that in the future we may discover that a quantitative description of these effects require crucial ingredients not considered here. As an example, see e.g. the strong modification of the critical field produced by inhomogeneities in the superconducting phase, arXiv:1609.09482. That work shows that B_c can become arbitrarily small when the condensate's phase changes across the nanowire width.

We hope that our experimental evidences and standard theoretical model presented will stimulate a deeper investigation of the topological superconductivity in complex real systems to prove or disprove our scenario. Until now, as argued in the manuscript, amongst all candidate energy scales in the problem, the smallest, and hence the most likely to be related to the observed field for the I_c anomaly, is the induced superconducting gap Δ , which is the critical field in the topological context.

We wish to thank the Referees for their positive comments on our paper. In the following we will provide the answers to the specific comments of Referee #3.

Referee #3

Question 1) The Referee states: *“The new material clarifies most of the issues raised in the previous round and provides reasonable explanations concerning some important limitations (e.g., why it is difficult to investigate a configuration involving a magnetic field parallel to the wire). Considering, as a whole, the experimental component of this work and the possible explanation based on a simple (but reasonable) model, I think that the manuscript should be considered for publication. My only recommendation would be to include a brief summary of the experimentally-relevant conditions that could generate deviations from the theoretical predictions based on the idealized model (e.g., finite size effects, electrostatic effects, etc.). It is important to clearly convey to the reader that some of the theoretical estimates (e.g., the critical field) are significantly different from the measured quantities and to provide possible causes for these (sometimes large) discrepancies.”*

Answer: We thank the Referee for his/her recommendation. We included in the summary of the amended version of the manuscript the following statement: *“We presented a model, based on topological transitions, that allows us to qualitatively explain all the observed phenomena. Quantitatively our theoretical prediction gives only a rough estimate of the transition field B_{crt} , still one order of magnitude higher than the experimental one B_{sw} . This discrepancy might be reduced by incorporating non-trivial but experimentally-relevant effects into the theory, including finite size in the NW, electrostatic effects, and scattering events.”*

We hope that Referee #3 is satisfied with our answers to her/his comments, and we believe that with this revision our manuscript can be published in Nature Communications.